# DISTRIBUTIONAL INVERSE REINFORCEMENT LEARNING

## ABSTRACT

We propose a distributional framework for offline Inverse Reinforcement Learning (IRL) that jointly models uncertainty over reward functions and full distributions of returns. Unlike conventional IRL approaches that recover a deterministic reward estimate or match only expected returns, our method captures richer structure in expert behavior, particularly in learning the reward distribution, by minimizing first-order stochastic dominance (FSD) violations and thus integrating distortion risk measures (DRMs) into policy learning, enabling the recovery of both reward distributions and distribution-aware policies. This formulation is well-suited for behavior analysis and risk-aware imitation learning. Theoretical analysis show that the algorithm converge with $\mathcal{O}(\varepsilon^{-2})$ iteration complexity. Empirical results on synthetic benchmarks, real-world neurobehavioral data, and MuJoCo control tasks demonstrate that our method recovers expressive reward representations and achieves state-of-the-art imitation performance.

## 1 INTRODUCTION

Inverse Reinforcement Learning (IRL) aims to infer an expert's underlying reward function and policy from observed trajectories collected under unknown dynamics. IRL has been successfully applied in diverse domains, including robotics (Vasquez et al., 2014; Wu et al., 2024a), animal behavior modeling (Ashwood et al., 2022; Ke et al., 2025), autonomous driving (Rosbach et al., 2019; Wu et al., 2020), and fine-tuning of large language models (Zeng et al., 2025). A pioneering work in this field, the Maximum Entropy IRL (MaxEntIRL) framework (Ziebart et al., 2008), formulates reward learning as a likelihood optimization problem and interprets expert policies as Boltzmann distributions over returns. Follow-up works have extended this framework to improve reward inference stability and generalization (Arora & Doshi, 2021; Garg et al., 2021; Zeng et al., 2022).

Despite these advances, most IRL methods assume that the expert's reward function is deterministic, thereby recovering only a point estimate, i.e., $r(s, a) \in \mathbb{R}$ for every state $s$ and action $a$. This assumption, however, limits expressiveness in real-world settings where reward signals are inherently stochastic. For instance, in robotic manipulation tasks involving deformable or fragile objects (Yin et al., 2021), contact uncertainty introduces reward variability for identical state-action pairs—variability that directly influences the learned policy's robustness and safety. Similarly, in neuroscience, dopaminergic neuron activity has been shown, as reward signals, to drive animal behavior via RL policies (Markowitz et al., 2023b). Yet, dopamine signals exhibit significant trial-to-trial variations, suggesting that behavior may arise from an underlying stochastic reward distribution. These challenges are further amplified in offline IRL settings, where interaction with the environment is unavailable and the algorithm must fully rely on fixed demonstrations.

These examples highlight that in many real-world scenarios, demonstrations may be generated under stochastic reward functions, i.e., $r(s, a)$ is a random variable. This motivates the need to go beyond point estimates and instead recover the full distribution of rewards. Prior works such as Bayesian IRL (BIRL) methods infer a posterior over reward parameters using Markov chain Monte Carlo (MCMC) (Ramachandran & Amir, 2007), Maximum a posteriori (MAP) estimation (Choi & Kim, 2011), or variational inference (Chan & van der Schaar, 2021), but primarily capture uncertainty over the parameters of a deterministic reward function. More importantly, BIRL still optimizes the expected return, following the MaxEntIRL framework, failing to exploit the richer structure present in the full return distribution induced by stochastic rewards. In other words, if reward learning in IRL is based solely on maximizing expected return, then the resulting policy is influenced only by

the mean and remains insensitive to the variance or higher-order moments of the reward. As a result, such an approach provides insufficient signal for accurately estimating the full reward distribution.

However, it remains unclear how to effectively learn reward distributions directly from expert demonstrations. Conventional MaxEntIRL fails to capture higher-order moments of the return, motivating the use of statistical distances between return distributions. Yet, such approaches introduce significant challenges for policy learning, the dual problem to reward inference, because most statistical distances couple the estimated return distribution with the (unknown) expert return distribution. This coupling exacerbates compounding errors and prevents leveraging established distributional RL techniques. Consequently, a principled framework is needed that enables reward distribution learning while simultaneously supporting return distribution estimation in the offline IRL setting.

To this end, we introduce *Distributional Inverse Reinforcement Learning* (DistIRL), a novel framework that explicitly models both the distributional nature of reward and the return. This allows us to capture stochasticity not only from transitions and policies but also from the reward function itself. Specifically, for reward learning, instead of matching expected returns as in MaxEntIRL, we propose to match the full return distribution using a First-order Stochastic Dominance (FSD) criterion. This allows us to capture not only the mean but also higher-order moments of the return distribution and thus capturing the full landscape of reward distributions, leading to a richer and more faithful estimate of the underlying reward structure. To the best of our knowledge, *this is the first work that learn the full distribution of the reward function in a principled manner.*

It is important to note that while our framework incorporates risk-sensitive policy learning, risk sensitivity primarily serves as a mechanism that enables robust reward distribution learning in the offline IRL setting. The connection is explained in detail in Sec. 4.2. Our contributions in this paper are summarized as follows:

(1) **Reward Distribution Learning.** We propose an intuitive framework for learning reward distributions in the offline IRL setting. With FSD objective emphasizing the match of the entire distribution, we are able to learning reward distributions beyond the first moment.

(2) **Distribution-aware Policy Learning.** Our algorithm learns the return distribution and recovers the distribution-aware policy, extending the modeling capability of IRL frameworks towards a broader range of behavior analyses and facilitating imitation learning in risk-sensitive scenarios.

(3) **Theoretical Analysis.** We develop rate of convergence analysis for the proposed algorithm for solving DistIRL, which shows that the algorithm converge with $\mathcal{O}(\varepsilon^{-2})$ iteration complexity.

(4) **Empirical Validation.** We demonstrate that our method recovers meaningful reward distributions on synthetic and real-world datasets, including neurobehavioral data (first-time studied for IRL). Our algorithm also achieves state-of-the-art performance on high-dimensional robotic control tasks in offline IRL settings.

## 2 RELATED WORK

**Inverse Reinforcement Learning**  Traditional offline IRL algorithms recover a reward function by matching expert feature expectations or maximizing an entropy-regularized likelihood. Apprenticeship learning (Abbeel & Ng, 2004) and MaxEntIRL (Ziebart et al., 2008; 2010) infer a deterministic reward whose induced policy reproduces expert behavior in expectation. Subsequent deep IRL variants incorporate neural network function approximators in the online setting (Ho & Ermon, 2016; Jeon et al., 2018; Wulfmeier et al., 2015; Ni et al., 2021; Garg et al., 2021; Zeng et al., 2022; Gleave & Toyer, 2022; Viano et al., 2021; Bloem & Bambos, 2014; Wu et al., 2024b; Zhan et al., 2024), where a subset of work using a variant of this framework, Maximum Casual Entropy IRL (MCE-IRL), emphasizing the casual relationship in its nature, in which the policy further interacts with the environment but still match only the expected return. As a result, these approaches cannot capture risk preferences or higher-order statistics of the reward distribution present in many real-world tasks. In addition, online IRL methods require interactive access to a simulator during training, which is unsuitable for offline settings where reproducing the environment is undesirable or infeasible, e.g. modeling mouse behavior in a maze. Finally, while recent work has explored risk-aware policy learning within the IRL framework (Singh et al., 2018; Lacotte et al., 2019; Cheng et al., 2023), these approaches still assume a deterministic reward model, failing to capture the stochasticity of rewards in many real-world problems. We show a detailed comparison of IRL methods across modeling assumptions in Appendix A.

**Bayesian Imitation Learning**  Bayesian IRL (BIRL) methods infer a posterior distribution over reward parameters to quantify uncertainty in reward estimation. Ramachandran and Amir (Ra-

machandran & Amir, 2007) introduces the first Bayesian IRL, using MCMC to sample from the reward posterior under a Boltzmann-rationality likelihood. Follow-up works use the same framework to handle larger state spaces and richer reward priors (Choi & Kim, 2011; Levine et al., 2011; Chan & van der Schaar, 2021; Li et al., 2023). Although these methods capture parameter uncertainty, they still rely on expected-return assumptions and do not exploit the full return distribution. Moreover, BIRL with a reward distribution fails to model continuous action spaces as obtaining the likelihood is computationally intractable for passing the gradient to the reward posterior. In this work, we propose a scalable algorithm framework for learning the full reward distributions.

**Distributional Reinforcement Learning**  DistRL extends classical value-based methods by modeling the full distribution of returns rather than only their expectation. Early work, such as Categorical DQN (C51) (Bellemare et al., 2017) and Quantile Regression DQN (QR-DQN) (Dabney et al., 2018b), demonstrates that learning a distributional critic improves stability and sample efficiency. More recent advances include Implicit Quantile Networks (IQN) (Dabney et al., 2018a), Implicit Q-Learning (Kostrikov et al., 2021), Multivariate Distribution RL (Wiltzer et al., 2024), and Diffusion Process for RL (Hansen-Estruch et al., 2023; Li et al., 2024). Note that DistRL still inherently maximizes the expected return. Risk-sensitive extensions (Lim & Malik, 2022; Schneider et al., 2024) that optimize risk measures like CVaR, show that one can directly shape policies by tailoring decisions to specific regions of the return distribution. While these methods are widely adopted in RL, the IRL counterparts (Lee et al., 2022; Karimi & Ebadzadeh, 2025) with a distributional critic are limited in scope. These methods use a distributional critic to model return distributions and extract expert policies, but still assume deterministic reward functions, and take on MaxEntIRL as the blueprint, i.e., matching the mean of the return distribution.

## 3 PRELIMINARIES

We model the environment as a discounted Markov Decision Process (MDP) $(\mathcal{S}, \mathcal{A}, P, r, \gamma)$, where $\mathcal{S}$ denotes the state space, $\mathcal{A}$ the action space, $P(s'|s, a)$ the transition kernel, and $\gamma \in [0, 1)$ the discount factor. The reward function is a (integrable) random variable $r : (\Omega, \mathcal{F}, \mathbb{P}) \to (\mathbb{R}, \mathcal{B}(\mathbb{R}))$, so that for each state–action pair $(s, a)$, the reward $r(s, a)$ induces a probability distribution over $(\mathbb{R}, \mathcal{B}(\mathbb{R}))$. A policy $\pi(a|s)$ generates a trajectory $(s_0, a_0, s_1, a_1, \ldots)$, and the associated (discounted) return is the random variable $Z^\pi = \sum_{t=0}^{\infty} \gamma^t r(s_t, a_t)$.

### 3.1 MAXIMUM ENTROPY INVERSE REINFORCEMENT LEARNING

Given demonstrations $\{(s_t, a_t)\}_{t \geq 1}$ collected by an unknown expert policy $\pi^E$, MaxEntIRL (Ziebart et al., 2008) aims to recover the unknown policy, and the corresponding reward function $r$ which the policy is optimized to. Specifically, we consider the following formulation (Ho & Ermon, 2016):

$$\max_\pi \min_r \mathbb{E}_{d^\pi}[r(s, a)] - \mathbb{E}_{d^{\pi^E}}[r(s, a)] + \mathcal{H}(\pi) + \psi(r), \tag{1}$$

where $\mathcal{H} := \mathbb{E}_{d^\pi}[-\log \pi(a|s)]$ denotes the entropy, and $\psi$ is a general convex regularizer. This formulation reduces to MaxEntIRL if $\psi = 0$. If $\psi = \mathrm{KL}(q(r)||p_0(r))$, it can be seen as a BIRL framework, since the optimal policy follows a Boltzmann distribution of the action-values[1].

## 4 DISTRIBUTIONAL INVERSE REINFORCEMENT LEARNING FRAMEWORK

In our model, we treat the reward as a *distribution* rather than a deterministic function. During optimization, the first two terms in Eq. 1, $\mathbb{E}_{d^\pi}[r(s, a)] - \mathbb{E}_{d^{\pi^E}}[r(s, a)]$, enforce *mean dominance*—that is, the learned reward should yield a higher expected return for the expert policy than for any arbitrary policy. At optimality, this difference becomes zero, indicating *mean matching* between expert and agent returns. However, if the reward is inherently a distribution, mean matching alone fails to capture the relationship between the expert's return distribution and the agent's in its entirety. This leads to a loss of higher-order information in the reward. To accurately model the full reward distribution, we must impose a *distributional form of dominance* during optimization, ensuring that the entire return distribution is aligned at optimality, not just the mean.

Let's consider a notion of order in term of the entire distributions.

**Definition 4.1** (First-Order Stochastic Dominance (FSD) (Hadar & Russell, 1969)). Let $X$ and $Y$ be real-valued integrable random variables with cumulative distribution functions $F_X$ and $F_Y$. We say that $X$ *first-order stochastically dominates* $Y$, written as $X \succeq_{\mathrm{FSD}} Y$, if $F_X(z) \leq F_Y(z), \forall z \in \mathbb{R}$.

---

[1]The Kullback-Leibler divergence is convex in its first argument when the second argument is fixed.

The concept of FSD is illustrated in Fig. 1. If we aim for $X \succeq_{\text{FSD}} Y$, then the shaded region indicates a violation of this condition. FSD has an equivalent definition relating to utility functions, which further implies mean dominance.

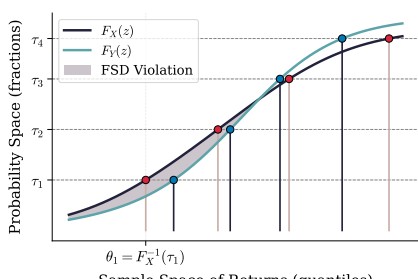

**Proposition 4.2** (Theorem 1-2 (Hadar & Russell, 1969)). *For real-valued $X$ and $Y$, the following are equivalent:*

1. $F_X(z) \leq F_Y(z)$ *for all* $z \in \mathbb{R}$.
2. $\mathbb{E}[u(X)] \geq \mathbb{E}[u(Y)]$ *for every non-decreasing utility function* $u : \mathbb{R} \to \mathbb{R}$.

*Corollary* 4.3 (Mean Dominance). If $X \succeq_{\text{FSD}} Y$, it follows that $\mathbb{E}[X] \geq \mathbb{E}[Y]$, as the identity utility $u(x) = x$ is non-decreasing.

Figure 1: Illustration of quantile functions and first-order stochastic dominance (FSD).

We model the reward as a conditional distribution, $r_t \sim q(\cdot|s_t, a_t)$, and define the random return for a trajectory $(s_0, a_0, \dots)$ sampled from policy $\pi$ as $Z^\pi = \sum_{t=0}^{\infty} \gamma^t r_t$. We now introduce the distributional counterpart to Eq. 1, the objective for distributional IRL, expressed as

$$\max_{\pi} \min_{r} \mathcal{L}(\pi, r) := \max_{\pi} \min_{r} \int_{-\infty}^{\infty} [F_{Z^E}(z) - F_{Z^\pi}(z)]_+ dz + \mathcal{H}(\pi) + \psi(r), \tag{2}$$

where $Z^E$ denotes the return distribution of the expert policy.

### 4.1 Learning Reward Distribution through Stochastic Dominance

From Eq. 2, the objective of the reward function is

$$\min_{r} \mathcal{L}_{\text{FSD}}(\pi, r) + \psi(r) = \min_{r} \int_{-\infty}^{\infty} [F_{Z^E}(z) - F_{Z^\pi}(z)]_+ dz + \psi(r). \tag{3}$$

This objective minimizes the violation of FSD, drawing inspiration from the Kolmogorov-Smirnov (K-S) test (Massey Jr, 1951). To model the reward distribution in a principled manner, we treat $\mathcal{L}_{\text{FSD}}(\pi, r)$ as an *energy function* that scores how compatible a proposed reward $r$ is with the expert demonstrations. In particular, we define a likelihood function over the expert demonstrations $\mathcal{D}$ using the Energy-Based Model (EBM) formulation (LeCun et al., 2006): $p(\mathcal{D}|r) \propto \exp(-\mathcal{L}_{\text{FSD}}(\pi, r))$, so that reward functions that yield small FSD violations are exponentially more likely under the expert data. This construction is natural here because FSD does not provide an explicit probabilistic model, but *does* provide a calibrated energy landscape that reflects goodness-of-fit. A more detailed discussion can be found in Appendix B.3.

We also introduce a *prior distribution* $p_0(r)$, which reflects our initial belief before observing any data. The goal is to infer the posterior distribution $p(r|\mathcal{D})$ using Bayes' rule. As direct inference under the EBM formulation is generally intractable, we adopt the variational inference framework (Blei et al., 2017) by introducing a *variational distribution* $q_\phi(r|s, a)$, parameterized by $\phi$, to approximate the posterior and optimize the *evidence lower bound (ELBO)*:

$$\text{ELBO} = \mathbb{E}_{q_\phi(r|s,a)}[\log p(\mathcal{D}|r)] - \text{KL}(q_\phi(r|s, a) \| p_0(r)). \tag{4}$$

Substituting the energy-based likelihood into the ELBO yields:

$$\min_{\phi} \mathcal{L}_r(\phi) := \min_{\phi} \mathbb{E}_{q_\phi(r|s,a)}[\mathcal{L}_{\text{FSD}}(\pi, r)] + \text{KL}(q_\phi(r|s, a) \| p_0(r)). \tag{5}$$

Notice the natural relationship between KL and $\psi$. Formally, we learn the reward distribution by solving Eq. 5. To compute the gradient of the first term, we apply the Inverse Transform Sampling technique (Devroye, 2006). We use the empirical quantile to approximate the quantile of the return. Specifically, using the change of variable formula, and the relation between CDF and quantile, we have

$$\int_{-\infty}^{\infty} [F_{Z^E}(z) - F_{Z^\pi}(z)]_+ dz = \int_0^1 \left[F_{Z^\pi}^{-1}(v) - F_{Z^E}^{-1}(v)\right]_+ dv. \tag{6}$$

We provide a short proof of the above relation in Appendix C.1. To approximate $F_\pi^{-1}$, we draw $N$ samples $\{z_n\}$ by Monte Carlo sampling $z_n = \sum_0^\infty \gamma^t r_t, r_t \sim q_\phi(\cdot|s_t, a_t)$, and form the empirical quantile using its order statistics $F_{Z^\pi}^{-1} \approx (z_{(-N)}, \dots, z_{(1)})$. As a result, minimizing $\mathcal{L}_r(\phi)$ generalizes the usual IRL objective of matching expected returns by aligning higher-order moments beyond matching the mean.

## 4.2 RISK-AWARE POLICY LEARNING

Once the inner minimization over $r$ yields a fixed reward distribution, the policy, parameterized by $\varphi$, is updated by maximizing the following objective:

$$\max_{\varphi} \mathcal{L}_{\pi}(\varphi) = \max_{\varphi} \int_0^1 [F_{Z^{\pi_{\varphi}}}^{-1}(v) - F_{Z^E}^{-1}(v)]_+ dv + \mathcal{H}(\pi_{\varphi}). \tag{7}$$

Let's define $\mathcal{I}(v) := \mathbb{1}_{F_{Z^{\pi_{\varphi}}}^{-1}(v) \geq F_{Z^E}^{-1}(v)}$. Fig. 1 shows that $\mathcal{I}(v)$ takes the value 1 in regions where FSD is violated (shaded area), and 0 otherwise. We then rewrite the objective in Eq. 7 as

$$\int_0^1 \left( F_{Z^{\pi_{\varphi}}}^{-1}(v) - F_{\pi^E}^{-1}(v) \right) \mathcal{I}(v) dv + \mathcal{H}(\pi_{\varphi}). \tag{8}$$

Note that the indicator function $\mathcal{I}$ depends on the current policy, the expert policy, and the quantile level $v$. Conceptually, $\mathcal{I}$ assigns weight only to regions of the return distribution where FSD is violated. The policy now aims to increase these FSD violations—encouraging the agent to obtain higher return samples in those regions. This leads to a maximization scheme that is inherently risk-aware, as it requires reasoning over the full return distribution rather than just its expectation.

Unfortunately, directly optimizing Eq. 7 is intractable, as the indicator function $\mathcal{I}$ is not observable during training. To address this, we take a broader perspective on risk-aware policy learning and propose replacing $\mathcal{I}(v)$ with a risk measure that retains the goal of encouraging risk-sensitive behavior while yielding a tractable objective. Furthermore, we show that the resulting surrogate objective provides a weaker form of optimality, but under certain conditions, it can theoretically achieve the same optimum as Eq. 7. To present our new objective, we need a few essential concepts.

**Definition 4.4** (Distortion function). A distortion function $\xi$ is a non-decreasing function $\xi : [0, 1] \to [0, 1]$ such that $\xi(0) = 0, \xi(1) = 1$.

**Definition 4.5** (Distortion Risk Measure (DRM) (Dhaene et al., 2012)). For an integrable random variable $X$, and a distortion function $\xi$, a Distortion Risk Measure $M_{\xi}$ is defined as

$$M_{\xi}(X) = \int_0^1 F_X^{-1}(v) d\tilde{\xi}(v), \tag{9}$$

where $\tilde{\xi} = 1 - \xi(1 - v) \geq 0$ is the dual distortion function.

Common examples of DRMs and distortion functions are listed in Table 1. These measures offer various ways to quantify risk based on the return distribution. Intuitively, when $\tilde{\xi}$ is concave, it places greater emphasis on lower returns, thereby encouraging risk-averse behavior. To induce risk-aware policies using distortion $\xi(v)$, we need to maximize the DRM defined in Eq. 9.

Table 1: Examples of distortion risk measures.

| Risk Measure | $\xi(v)$ | Interpretation |
|---|---|---|
| CVaR$_{\alpha}$ | $\min(v/\alpha, 1)$ | Average of worst $\alpha$-fraction of outcomes |
| Wang's Transform | $\Phi(\Phi^{-1}(v) + \lambda)$ | $\lambda > 0$ implies risk-aversion, $\lambda < 0$ risk-seeking |

Building on the above definitions, we propose replacing $\mathcal{I}(v)$ with $\tilde{\xi}(v)$ in Eq. 8, resulting in:

$$\max_{\varphi} \int_0^1 \left( F_{Z^{\pi}}^{-1}(v) - F_{Z^E}^{-1}(v) \right) d\tilde{\xi}(v) + \mathcal{H}(\pi) = \max_{\varphi} \int_0^1 F_{Z^{\pi}}^{-1}(v) d\tilde{\xi}(v) + \mathcal{H}(\pi). \tag{10}$$

The equality is obtained as the expert policy does not depend on $\varphi$. We denote the final objective as

$$\max_{\varphi} \mathcal{L}_{\pi}(\varphi) := \max_{\varphi} M_{\xi}(Z^{\pi_{\varphi}}) + \mathcal{H}(\pi_{\varphi}) = \max_{\varphi} \int_0^1 F_{Z^{\pi_{\varphi}}}^{-1}(v) d\tilde{\xi}(v) + \mathcal{H}(\pi_{\varphi}), \tag{11}$$

where $M_{\xi}$ is a chosen DRM with a distortion function $\xi$.

*Relation to Eq. 7.* Additionally, we know that $X \succeq_{\text{FSD}} Y \Rightarrow M_{\xi}(X) \geq M_{\xi}(Y)$ (Sereda et al., 2010). Then naturally one wonders what's the sufficient condition for FSD? We observe that the converse implication requires a stronger condition.

**Proposition 4.6.** $M_{\xi}(X) \geq M_{\xi}(Y)$ *for every distortion function $\xi$ implies $X \succeq_{\text{FSD}} Y$.*

The proof is straightforward by observing that $M_\xi(X) - M_\xi(Y) = \int_0^1 (F_X^{-1}(v) - F_Y^{-1}(v)) d\tilde{\xi}(v)$ and the fact that $\tilde{\xi}(v) \geq 0$. We present a short proof in Appendix C.1. This implies that if we solve $\max_{\pi_\varphi} \int_0^1 \left( F_{Z^{\pi_\varphi}}^{-1}(v) - F_E^{-1}(v) \right) d\tilde{\xi}(v) + \mathcal{H}(\pi_\varphi)$ for every distortion function, we obtain the solution to Eq. 7. However, since optimizing over all utility conditions is intractable, our proposed objective serves as an approximation using a specific DRM. Nonetheless, under the conditions of the proposition, this surrogate objective can theoretically achieve the same optimality as Eq. 7.

## 4.3 Practical Algorithm

---

**Algorithm 1:** A DistIRL method with FSD objective

---

**Input:** Expert data $\mathcal{D} = \{(s_t^E, a_t^E)\}$, prior $p_0(r)$, risk measure $\xi$, step sizes $\eta^\theta, \eta^\varphi, \eta^\phi$
**Output:** Reward distribution $q_\phi(r|s, a)$; policy $\pi_\varphi(a|s)$

1 Initialize parameters of reward network $\phi$, policy $\varphi$, and critic $\theta$;
2 **for** $k = 1$ **to** $K$ **do**
3      Sample a mini-batch $\{(s_t^E, a_t^E)\}$ from $\mathcal{D}$;
4      **foreach** $(s_t^E, a_t^E)$ *in mini-batch* **do**
5          For each $s_t^E$, sample $a_t \sim \pi_\varphi(\cdot|s_t^E), r_t \sim q_\phi(\cdot|s_t^E, a_t), r_t^E \sim q_\phi(\cdot|s_t^E, a_t^E)$;
6      Compute return samples $Z^{\pi_k}, Z^E$;
7      Critic update via quantile regression (Eq. 20): $\theta_{k+1} \leftarrow \theta_k - \eta^\theta \nabla \mathcal{L}_{QR}(\theta_k)$;
8      Policy update with distortion risk measure (Eq. 11): $\varphi_{k+1} \leftarrow \varphi_k - \eta^\varphi \nabla \mathcal{L}_\pi(\varphi_k)$;
9      Reward distribution update via FSD loss (Eq. 5): $\phi_{k+1} \leftarrow \phi_k - \eta^\phi \nabla \mathcal{L}_r(\phi_k)$.

---

To enable tractable and expressive modeling of reward uncertainty, we parameterize the reward distribution $q_\phi(r|s, a)$, for example, using Azzalini's skew-normal distribution (Azzalini & Valle, 1996): $q_\phi(r|s, a) = \mathcal{SN}(\mu_\phi(s, a), \sigma_\phi^2(s, a); \alpha_\phi(s, a))$, where the mean $\mu_\phi(s, a)$, standard deviation $\sigma_\phi(s, a)$ and the skew parameter $\alpha_\phi(s, a))$ are outputs of a neural network with parameters $\phi$. This choice allows for efficient sampling and computing regularization when using a standard normal prior. During training, for each state-action pair, we sample rewards $r_t \sim q_\phi(\cdot|s_t, a_t)$ to construct return samples for both the expert and the current policy.

Note that the choice of prior depends heavily on the task domain and the type of variability we expect in the reward signal. For example, skew-normal distributions can capture asymmetric reward uncertainty in tasks with systematic biases (e.g., contact-rich manipulation), whereas heavy-tailed priors may be more suitable when outliers or rare but significant events dominate the return structure. In contrast, the broader statistical learning community often defaults to Gaussian priors, primarily because of their analytical tractability, conjugacy with many likelihood models, and well-understood concentration properties. That said, DistIRL does not rely on a fixed distributional assumption. Any parameterized distribution $p_\theta$ whose log-density or quantile function is differentiable in $\theta$ is compatible with our framework, since the algorithm requires only gradient updates for learning.

To estimate the spectral risk measure $M_\xi(Z^\pi)$ for the policy, we follow an offline approach: we use states $s_t$ drawn from the expert demonstration dataset, but sample actions $a_t^\pi \sim \pi_\theta(\cdot|s_t)$ from the current policy, and a reward $r_t \sim q_\phi(\cdot|s_t, a_t)$. Then we compute the return $Z^\pi$ by taking the sum. For policy update, we first learn the critic by Off-policy Evaluation (OPE) (Sutton et al., 1998) on $(s_t, a_t, r_t, s_{t+1}, a_{t+1}^\pi)$ where we use Quantile Regression with the Quantile Huber loss $\mathcal{L}_{QR}$ as in Eq. 20. We then update the risk-aware policy by solving $\min_\pi \mathrm{KL}\left(\pi(\cdot|s) \,\middle\|\, \frac{1}{Z} \exp\left\{M_\xi(Z^\pi(\cdot|s))\right\}\right)$, which corresponds to the KKT solution to Eq. 7, as originally introduced by Ziebart et al. (2008). We summarize the full procedure in Alg. 1.

## 5 Theoretical Results

In this section, we provide theoretical analysis of the algorithm proposed above. In particular, this analysis framework assume that we know the exact DRM for solving the policy update in Eq. 7. First, we introduce several regularity assumptions, the necessity of which is detail in the appendix C.2.

**Assumption 5.1.** *There exists $R_{\max} < \infty$ such that*

$$|q_\phi(s, a)| \leq R_{\max} \quad \text{almost surely for all } (s, a, \phi). \tag{12}$$

**Assumption 5.2.** *For every $(s, a)$ and all $\phi_1, \phi_2 \in \mathbb{R}^d$, the reward laws satisfy*

$$W_\infty\big(q_{\phi_1}(\cdot|s,a),\, q_{\phi_2}(\cdot|s,a)\big) \;\leq\; L_R \,\|\phi_1 - \phi_2\|, \tag{13}$$

*where $W_\infty$ denotes the Wasserstein inifity distance. Equivalently, one can couple $q_{\phi_1}(s, a)$ and $q_{\phi_2}(s, a)$ such that $|q_{\phi_1}(s, a) - q_{\phi_2}(s, a)| \;\leq\; L_R \,\|\phi_1 - \phi_2\|$ almost surely.*

We use the following assumption on a given DRM. In fact, all DRMs satisfy the following properties.

**Assumption 5.3.** *For each state-action pair $s \in \mathcal{S}, a \in \mathcal{A}$, the one-step distortion risk measure $M_\xi(\cdot|s, a)$ is*

1. monotone: *$X \leq Y$ a.s. implies $M_\xi(X|s, a) \leq M_\xi(Y|s, a)$;*

2. translation-equivariant: *$M_\xi(X + c|s, a) = M_\xi(X|s, a) + c$ for all $c \in \mathbb{R}$;*

3. 1-Lipschitz in $\|\cdot\|_\infty$: *for all bounded random variables $X, Y$,*

$$\big|M_\xi(X|s,a) - M_\xi(Y|s,a)\big| \;\leq\; \|X - Y\|_\infty. \tag{14}$$

First we wish to show that the critic under a given DRM will converge in the average sense:

**Theorem 5.4.** *Assume assumptions 5.1-5.3 hold. Let $E_k = \big\|Q^\xi_{\phi_k,\pi_k} - Q^\xi_{\phi_k,\pi^\star_{\phi_k}}\big\|_\infty$. Assume the reward update satisfies Assumption C.9, with stepsizes $\eta_k = \eta = \eta_0 K^{-\sigma}$, $\eta_0 > 0$, and $\sigma \in (0, 1)$. Then running the DistIRL algorithm $K$ steps, we have*

$$\frac{1}{K} \sum_{k=1}^{K} E_k \;=\; \mathcal{O}(K^{-1}) \,+\, \mathcal{O}(K^{-\sigma}). \tag{15}$$

Here we assume that we can get the exact $\mathcal{I}$ function when solving the policy optimization problem. Then we can also get a policy bound:

**Theorem 5.5.** *For each $k$, define the learned and DRM-optimal policies induced by the current Q-functions:*

$$\pi_k(\cdot|s) \propto \exp\big(Q^\xi_{\phi_k,\pi_k}(s,\cdot)\big), \pi^\star_{\phi_k}(\cdot|s) \propto \exp\big(Q^\xi_{\phi_k,\pi^\star_{\phi_k}}(s,\cdot)\big). \tag{16}$$

*Then running the DistIRL algorithm $K$ steps, we have*

$$\frac{1}{K} \sum_{k=1}^{K} \big\|\log \pi_k - \log \pi^\star_{\phi_k}\big\|_\infty \;=\; \mathcal{O}(K^{-1}) + \mathcal{O}(K^{-\sigma}). \tag{17}$$

Finally, we can get a rate of convergence towards a first-order stationary point:

**Theorem 5.6.** *Suppose Assumptions 5.1, 5.2, C.9, and C.11 hold. Let $\eta_k = \eta_0 k^{-\sigma}$ with $\eta_0 > 0$ and $\sigma \in (0, 1)$, and assume $\mathcal{L}_r$ is bounded below on $\Phi$. Then there exists $C > 0$ such that*

$$\frac{1}{K} \sum_{k=0}^{K-1} \mathbb{E}\big[\|\nabla\mathcal{L}_r(\phi_k)\|^2\big] \;=\; \mathcal{O}\big(K^{-1}\big) + \mathcal{O}\big(K^{-\sigma}\big) + \mathcal{O}\big(K^{-1+\sigma}\big), \tag{18}$$

In particular, picking $\sigma = 1/2$, we obtain a $\mathcal{O}(\epsilon^{-2})$ iteration bound on the algorithm.

## 6 EXPERIMENT

### 6.1 GRIDWORLD

We begin with a $5 \times 5$ gridworld environment where the agent is trained to navigate from the starting state $(2, 0)$ (left-center) to rewarding goal locations. Two high-reward states are placed at $(0, 4)$ (top-right) and $(4, 4)$ (bottom-right), with the top-right reward modeled as a stochastic outcome drawn from $\mathcal{N}(1, 1)$. The first column of Fig. 2 illustrates the ground-truth reward mean and variance.

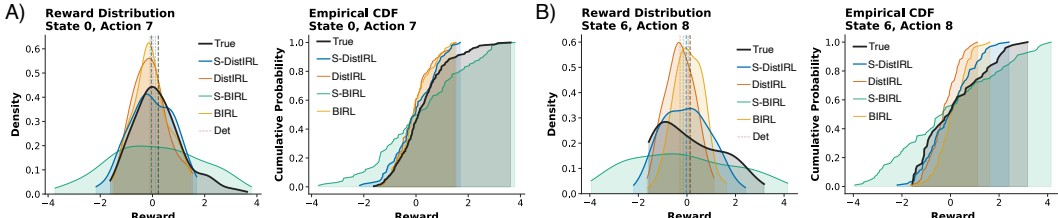

Figure 3: Learned reward distribution versus recorded dopamine signals and their empirical CDFs.

This setup mimics an animal exploring an arena with two reward ports. In such compact environments, animals often display risk-averse behavior, i.e., avoiding locations where rewards have previously failed to appear (Mobbs et al., 2018; Daw et al., 2006). To model this, we collect 10 trajectories from a risk-averse agent trained under stochastic rewards. In 9 out of 10 episodes, the agent chooses the more reliable bottom-right goal. We then apply our DistIRL method to recover the full reward distribution. As shown in Fig. 2, using a symmetric Gaussian reward estimator combined with risk-averse policy learning, our approach not only identifies both high-reward states but also captures the variance at the top-right goal.



Figure 2: Inferring reward mean and variance in the gridworld example with 10 demonstrations.

This highlights the model's ability to infer higher-order moments of the reward from expert demos.

As a baseline, we evaluate Bayesian IRL (BIRL) (Chan & van der Schaar, 2021; Mandyam et al., 2023; Bajgar et al., 2024). BIRL is a widely used framework that assumes a reward distribution but learns it by matching only the mean, without capturing the full distributional structure. We select BIRL because it is the method most comparable to ours in its ability to recover a reward distribution. BIRL reasonably recovers the mean reward but produces spurious high estimates in the lower-left corner. Furthermore, it fails to capture reward variance, emphasizing the need to enforce distance over the full distribution. Simply specifying a reward distribution, without integrating distribution-aware learning, fails to capture the true variance of the rewards.

## 6.2 MOUSE SPONTANEOUS BEHAVIOR

We apply our framework to a neuroscience dataset in which mice freely explore an arena without explicit rewards (Markowitz et al., 2023a). Behavior was recorded using a depth camera, and the raw trajectories were converted into sequences of discrete syllables (e.g., grooming, sniffing). We model these trajectories with an MDP, treating each syllable as a state and the next syllable as the action, yielding ten states and ten actions. In total, we analyzed 159 such state-action sequences. The dataset also includes a time-aligned one-dimensional trace of dopamine fluctuations from the dorsolateral striatum. Prior work (Markowitz et al., 2023a) showed that using dopamine as a reward enabled a simulated RL agent to reproduce observed transitions, suggesting IRL should recover a reward pattern resembling dopamine. Since dopamine varies even within the same state-action pair, the prior study used only its mean for simplicity. Here, we compare rewards learned under deterministic vs. distributional assumptions to assess how well they capture both the mean and the full distribution of dopamine signals.

We use both Azzalini's skew-normal distribution (denoted "S-") and the symmetric Gaussian as reward models for both DistIRL and BIRL. Fig. 3A) and B) show two example state-action pairs, illustrating the true dopamine fluctuation distribution alongside the estimated reward distributions from four methods. The assumption of a parameterized reward distribution is motivated by prior findings in computational neuroscience: dopamine-related reward signals in rodents are well known to exhibit asymmetric, left-skewed variability. For this reason, we chose a skew-normal family, which captures exactly this type of asymmetric structure while remaining interpretable. For each case, we display both the probability density function and the CDF, along with the corresponding means. Deterministic rewards (Det) are shown as pink dashed lines in the density plots. Among all methods, S-DistIRL most accurately recovers the shape of the dopamine distribution, which is often right-skewed and multimodal. Its estimated mean also closely matches both the true mean and the deterministic estimate.

We also quantify the similarity between estimated rewards and actual dopamine distributions. In Fig. 4A), we report the correlation between the mean of dopamine fluctuations and the mean of the estimated reward across all mice and trajectories. Deterministic reward models yield moderate correlation, while DistIRL improves upon this, with S-DistIRL achieving the highest correlation overall. This finding indicates that incorporating full reward

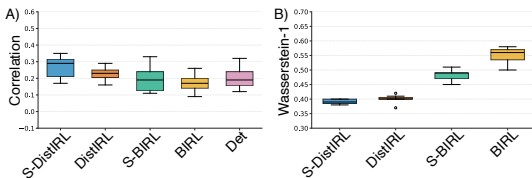

Figure 4: Left: Pearson correlation of the reward mean and dopamine level. Right: W-1 loss between learned distribution and dopamine level.

distributions, using suitable skewed distributional models, is essential for IRL to capture biologically meaningful reward signals. Fig. 4B) shows that, compared to BIRL, S-DistIRL also achieves a lower Wasserstein-1 distance between the estimated reward distribution and the actual dopamine distribution, indicating better alignment of the shape. Taken together, both qualitative examples and quantitative metrics support that modeling skewed reward distributions significantly enhances the ability to track dopamine fluctuations.

This is a scientifically interesting result showing that we can infer the reward structure directly from behavior data. While it is known that dopamine neurons encode reward-related signals (Schultz et al., 1997; Markowitz et al., 2023a), this is the first demonstration that not only is there a nontrivial correlation between the inferred and measured mean rewards (with a correlation around 0.3), but also that the full reward distribution recovered from behavior reasonably resembles the distribution of dopamine fluctuations. This suggests that detailed features of neuromodulatory signals, such as the variability in dopamine release, can be decoded from behavior alone, highlighting the potential of inverse modeling to uncover internal motivational states and their neural substrates.

### 6.3 MuJoCo Benchmarks

**Risk-sensitive D4RL.** In earlier experiments, we applied DistIRL to discrete state-action MDPs and compared it with BIRL. Here we extend the study to continuous MDPs to demonstrate DistIRL's scalability and generalizability. We evaluate our method on Risk-sensitive D4RL benchmarks, following the reward formulations introduced in recent robustness studies (Urpí et al., 2021). Specifically, the reward functions incorporate stochastic penalties triggered by safety-related conditions:
(1) **Half-Cheetah:** $R_t(s,a) = \bar{r}_t(s,a) - 70\mathbb{I}_{\nu>\bar{\nu}} \cdot \mathcal{B}_{0.1}$, where $\bar{r}_t(s,a)$ is the environment reward, $\nu$ is the forward velocity, and $\bar{\nu}$ is a velocity threshold ($\bar{\nu} = 4$ for the medium variant and $\bar{\nu} = 10$ for the easy variant). This penalty models rare but catastrophic robot failures at high speed.
(2) **Walker2D/Hopper:** $R_t(s,a) = \bar{r}_t(s,a) - p\mathbb{I}_{|\theta|>\bar{\theta}} \cdot \mathcal{B}_{0.1}$, where $\bar{r}_t(s,a)$ is the environment reward, $\theta$ is the pitch angle, $\bar{\theta}$ is a task-dependent threshold (0.5 for Walker2D-M/E and 0.1 for Hopper-M/E), and $p$ is the penalty magnitude (30 for Walker2D and 50 for Hopper).

We train expert agents on these stochastic reward formulations using Risk-averse Distributional SAC, a variant of DSAC (Duan et al., 2021) with CVaR objective, and collect 10 demonstration trajectories. We then evaluate DistIRL against several state-of-the-art baselines. Results are averaged over 5 random seeds. We use a standard normal as the prior due to its general applicability, in the setting of not knowing the underlying true reward distribution.

Table 2 shows that our method consistently outperforms other **offline** IRL baselines under stochastic reward settings. For reward parameterization, we use the Gaussian distribution (denoted as **DistIRL**) and quantile function (denoted as **DistIRL-qtr**, short for QuanTile Reward). Notice popular online methods such as GAIL (Ho & Ermon, 2016) are not applicable in this setting. **Offline ML-IRL** (Zeng et al., 2023) is a model-based MaxEntIRL method that relies on a separately trained transition model using additional non-expert data. Its poor performance here is expected: the transition model was pretrained under risk-neutral rewards and does not align with the new expert data generated under risk-sensitive objectives, leading to severe distribution mismatch. **ValueDICE** (Kostrikov et al., 2019), a model-free offline MaxEntIRL baseline, also underperforms since it optimizes with respect to expected risk-neutral returns, while our experts follow risk-averse behavior. **Behavior Cloning (BC)** achieves moderately strong results, as it simply mimics the demonstrated actions without explicitly optimizing for either risk-neutral or risk-sensitive objectives. However, its performance is limited as the model overfit the limited demonstration data.

To further validate the fidelity of our inferred return distributions from DistIRL and compare with the BIRL framework that only matches the mean, we collect 200 trajectories and sample its learned return distribution for each learned policy, plot against the expert's return distribution in Fig. 5.

Table 2: Performance averaged over 5 seeds on Risk-sensitive D4RL.

| Environment | DistIRL (ours) | DistIRL-qrt (ours) | Offline ML-IRL | ValueDICE | BC | Expert |
|---|---|---|---|---|---|---|
| HalfCheetah | **3469 ± 59** | 3294 ± 172 | 826 ± 231 | 1259 ± 78 | 2828 ± 281 | 3540 ± 44 |
| Hopper | **886 ± 1** | 747 ± 79 | 192 ± 56 | 260 ± 10 | 346 ± 1 | 892 ± 3 |
| Walker2d | **1526 ± 148** | 1211 ± 182 | 240 ± 50 | 798 ± 311 | 1321 ± 26 | 1478 ± 200 |

This shows that DistIRL's reward and policy model better align with the expert. We also report a Pearson correlation coefficient of 0.92 between the mean estimated by DistIRL and the mean of the true return. This indicates strong agreement and demonstrates that our inferred reward is an accurate proxy for the true reward model. A further examination of the return distribution and its higher-order moments can be found in Appendix F.

Additionally, the competitive results of quantile-based reward parameterization open the opportunity to use a broad range of parametric families, including diffusion models, and we leave this driection as a future extension of this work.

**Risk-neutral D4RL.** We also test our algorithm in conventional deterministic reward settings using D4RL's `medium-expert` trajectories (Fu et al., 2020). Table 3 shows our method achieves competitive or superior performance even without tailoring to deterministic assumptions, underscoring the generality of DistIRL. We want to emphasize that Offline ML-IRL requires additional data[2].

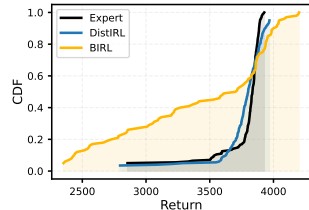

Figure 5: Return distributions comparison in HalfCheetah.

**Ablation studies.** We evaluate the contribution of different design choices by ablating our model under the HalfCheetah setting with right-skewed normal ($\mathcal{SN}_\eta, \eta > 0$) stochastic rewards and risk-averse expert policy, indicating the expert prefers conservative actions that yield more consistent rewards. Variants include: **Dis/Det**: Distributional or Deterministic rewards; **QR/TD**: Quantile Regression or TD-based critic; **FSD/Mean**: FSD loss or Mean matching. As shown in Table 4, which scales the performance between worst and best, using distributional rewards with FSD loss significantly outperforms mean-matching alternatives. Additionally, deterministic TD-learning with mean-matching (**Det-TD-Mean**) underperforms in learning risk-averse policies due to a lack of distributional supervision. This confirms the effectiveness of FSD-based reward learning and risk-sensitive policy optimization. Note that the BIRL framework aligns with our **Dis-TD-Mean** configuration; RIZE (Karimi & Ebadzadeh, 2025) aligns with **Det-Qt-Mean**, which performs the worst; **Det-TD-Mean** aligns with ValueDice but with an explicit reward estimation. Thus, in this ablation study, we treat them as a specific setting within DistIRL when benchmarking against other approaches.

Additionally, we conduct ablation studies on the choice of DRM in Appendix E.1, showing that DistIRL is not sensitive to specific DRM as long as we don't deviate too far from the underlying risk preference of the expert data. We also conduct experiments on the number of trajectories for the risk-sensitive D4RL dataset in Appendix E.2, which show that DistIRL is sufficiently robust in a low-data regime, indicating that our approach is indeed computationally attractive.

Table 3: Performance on deterministic reward settings (D4RL).

| Environment | DistIRL (Ours) | Offline ML-IRL | ValueDICE | BC | Expert |
|---|---|---|---|---|---|
| HalfCheetah | 7779 ± 228 | **11231 ± 585** | 4935 ± 2836 | 623 ± 56 | 12175 ± 91 |
| Hopper | **3411 ± 42** | 3347 ± 238 | 3073 ± 539 | 3236 ± 46 | 3512 ± 22 |
| Walker2d | **4570 ± 305** | 4201 ± 638 | 3191 ± 1888 | 2822 ± 979 | 5384 ± 52 |

Table 4: Ablation study on model setting. Performance scaled for clarity.

| DistIRL (Ours) | Dis-Qt-Mean | Det-Qt-Mean | Dis-TD-FSD | Dis-TD-Mean | Det-TD-Mean |
|---|---|---|---|---|---|
| **1.0 ± 0.02** | 0.22 ± 0.02 | 0.00 ± 0.01 | 0.67 ± 0.31 | 0.33 ± 0.01 | 0.22 ± 0.00 |

## 7 CONCLUSION

We introduce a distributional framework for inverse reinforcement learning that jointly models reward uncertainty and return distributions. Our method enables risk-aware policy learning and accurate inference of high-order structure in demonstrations. We validate the framework on stochastic control tasks, deterministic settings, and real neural datasets, demonstrating state-of-the-art performance and strong generalization across domains.

---

[2]For HalfCheetah, with the same amount of data as Offline ML-IRL, DistIRL can reach $11239 ± 539$.

## ETHICS STATEMENT

IRL enables powerful tools for understanding behavior, with positive applications in neuroscience, animal modeling, and AI alignment. However, it also raises ethical concerns. IRL could be misused in military settings to model or mimic adversarial behavior, or in surveillance contexts to infer personal goals without consent, posing risks to privacy and autonomy. These concerns highlight the need for careful oversight and responsible deployment.

## REPRODUCIBILITY STATEMENT

We list parameter choice in Table. 6. The implementation will be made publicly available following the paper decision.

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

## A    RELATED WORK COMPARISON

Table 5: Comparison of IRL methods under various settings

| Reference | Model reward dist.? | Infer risk aware policy? | Recover reward dist.? | Learn return dist.? |
|---|---|---|---|---|
| (Wulfmeier et al., 2015; Ziebart et al., 2008) (Garg et al., 2021; Ni et al., 2021) (Zeng et al., 2022; 2023; Wei et al., 2023) | ✗ | ✗ | ✗ | ✗ |
| (Ramachandran & Amir, 2007; Choi & Kim, 2011) (Chan & van der Schaar, 2021; Lee et al., 2022) | ✓ | ✗ | ✗ | ✗ |
| (Karimi & Ebadzadeh, 2025) | ✗ | ✗ | ✗ | ✓ |
| (Singh et al., 2018; Lacotte et al., 2019) (Cheng et al., 2023) | ✗ | ✓ | ✗ | ✗ |
| This work | ✓ | ✓ | ✓ | ✓ |

In Table A, we compare DistIRL with existing IRL methods along four key dimensions. The first column, *Model reward distribution*, asks whether a method explicitly represents the reward as a random variable rather than as a fixed deterministic function. For example, Bayesian IRL methods place a prior over reward parameters, thereby modeling uncertainty, but they do not recover the actual shape of the underlying distribution. This is distinct from *Recover reward distribution*, which requires learning the full distribution of rewards themselves, including higher-order statistics such as variance and skewness, rather than just a posterior over parameters.

The third column, *Infer risk-aware policy*, evaluates whether a method incorporates risk measures into policy inference. Methods in this category optimize beyond expected return, often capturing aversion or preference to variability in outcomes. The final column, *Learn return distribution*, indicates whether a method leverages distributional reinforcement learning (DistRL) techniques to estimate the full distribution of returns, rather than only their expectation. Unlike reward distributions, which describe stochasticity at the immediate reward level, return distributions capture the cumulative effect of randomness from rewards, transitions, and policies over trajectories.

As shown in the table, most prior IRL methods either assume deterministic rewards or restrict themselves to expectation-based inference. In contrast, DistIRL is the first framework that simultaneously models stochastic rewards, learns full reward distributions, integrates distributional return estimation, and supports risk-aware policy learning, thereby unifying these capabilities in a principled way.

## B    EXTENDED PRELIMINARIES

The state-value and action-value functions under $\pi$ are defined as

$$V^\pi(s) = \mathbb{E}\big[Z^\pi | s_t = s\big], \qquad Q^\pi(s,a) = \mathbb{E}\big[Z^\pi | s_t = s, a_t = a\big].$$

They satisfy the Bellman equations

$$V^\pi(s) = \mathbb{E}_{a\sim\pi, s'\sim P}\left[r(s,a) + \gamma V^\pi(s')\right], \quad Q^\pi(s,a) = \mathbb{E}_{s'\sim P}\big[r(s,a) + \gamma\,\mathbb{E}_{a'\sim\pi}[Q^\pi(s',a')]\big].$$

We also define the *occupancy measure* of $\pi$ as $d^\pi(s,a) = (1-\gamma)\sum_{t=0}^\infty \gamma^t \Pr(s_t = s)\,\pi(a|s)$, which satisfies $\sum_{s,a} d^\pi(s,a) = 1$ and characterizes the long-run state-action visitation probability.

### B.1    DISTRIBUTIONAL RL AND RISK-SENSITIVE CONTROL

Rather than estimating only $\mathbb{E}[Z^\pi]$, distributional RL models the entire return distribution that obeys the *distributional Bellman operator* $\mathcal{T}^\pi$ (Bellemare et al., 2017):

$$Z^\pi(s,a) = \sum_{t=0}^\infty \gamma^t\, r(s_t, a_t),$$

$$\mathcal{T}^\pi Z(s,a) \overset{\mathrm{D}}{=} r(s,a) + \gamma\, Z\big(s', \pi(s')\big),$$

where $V := \overset{\mathrm{D}}{=} U$ denotes equality of probability laws, indicating random variables $\{V, U\}$ are distributed according to the same law. A popular parameterization uses quantile regression: one approximates $Z^\pi(s,a)$ by $N$ quantiles $\boldsymbol{\theta}(s,a) = [\theta_1(s,a), ..., \theta_N(s,a)] : \mathcal{S} \times \mathcal{A} \to \mathbb{R}^N$ at fractions (quantile levels) $\tau_i = i/N$, for $i = 1, \ldots, N$. In other words, the quantile distribution of $Z^\pi(s,a)$ is represented a uniform probability distribution supported on $\{\theta_i(s,a)\}_{i=1}^N$: $Z^\pi(s,a) = \frac{1}{N} \sum_{i=0}^N \delta_{\theta_i}(s,a)$ where $\delta_{\theta_i}$ denotes a Dirac at $\theta_i$. An example of quantile functions is illustrated in Fig. 1, with $\theta$ and $\tau$ indicated.

To update the critic, instead of formulating the TD error, one can minimize the quantile Huber loss (Dabney et al., 2018b) with threshold $\kappa > 0$:

$$\rho_\tau^\kappa(\delta) = \big|\tau - \mathbf{1}\{\delta < 0\}\big|\, H_\kappa(\delta),\, H_\kappa(\delta) = \begin{cases} \frac{1}{2}\,\delta^2, & |\delta| \le \kappa, \\ \kappa\,|\delta|\, -\, \frac{1}{2}\,\kappa^2, & |\delta| > \kappa. \end{cases} \tag{19}$$

In distributional RL with $N$ quantile fractions $\{\tau_i\}$, the loss for the critic is defined as

$$\min_\theta \mathcal{L}_{\mathrm{QR}}(\theta) = \min_\theta \frac{1}{N} \sum_{i=1}^N \sum_{j=1}^N \rho_{\tau_i}\left(\delta_{ij}\right), \delta_{ij} = r + \gamma\,\theta_j(s',a') - \theta_i(s,a). \tag{20}$$

Once the return distribution is learned, one can optimize risk measures $M$, e.g. Conditional Value at Risk (CVaR) (Rockafellar et al., 2000), by maximizing $\mathrm{CVaR}\big(Z^\pi\big)$ rather than $\mathbb{E}[Z^\pi]$, yielding risk-sensitive policies.

**Deterministic reward as a special case.** If $q(\cdot \mid s,a)$ is a point mass at some value $r(s,a)$ for every $(s,a)$, then we recover the usual deterministic reward setting. Thus, our framework strictly generalizes standard IRL.

**Why distributions matter.** If the reward is inherently stochastic (for example, due to noisy human judgments), matching only the *mean* reward or mean return is not enough to capture the full behavior. Two policies can have the same expected return but very different risk profiles. This motivates working with the full return distribution $Z^\pi$, not just its expectation.

### B.2 First-Order Stochastic Dominance (FSD)

We now recall first-order stochastic dominance, which provides a way to compare entire distributions, not just means or variances.

**Definition B.1** (First-order stochastic dominance). Let $X$ and $Y$ be real-valued integrable random variables with cumulative distribution functions $F_X$ and $F_Y$. We say that $X$ *first-order stochastically dominates* $Y$, written $X \succeq_{\mathrm{FSD}} Y$, if

$$F_X(z) \le F_Y(z) \quad \text{for all } z \in \mathbb{R}.$$

Intuitively, $X \succeq_{\mathrm{FSD}} Y$ means that $X$ tends to take larger values than $Y$: for every threshold $z$, the probability that $X$ falls below $z$ is no larger than the probability that $Y$ does. Graphically, the CDF of $X$ lies everywhere *below* the CDF of $Y$.

**Connection to utilities and mean dominance.** A classical result states that $X \succeq_{\mathrm{FSD}} Y$ if and only if

$$\mathbb{E}[u(X)] \ge \mathbb{E}[u(Y)]$$

for every non-decreasing utility function $u$. In particular, taking $u(x) = x$, we get

$$\mathbb{E}[X] \ge \mathbb{E}[Y],$$

so FSD implies mean dominance. However, the converse is false: matching or exceeding the mean does *not* guarantee FSD.

**FSD in our context.** In our framework, we would like the return distribution of the expert policy, $Z^E$, to dominate that of any learned policy $Z^\pi$, or vice versa depending on the formulation. This is a strong requirement and is typically hard to enforce directly during learning. Our approach therefore designs an objective that *penalizes violations of FSD* and then turns this objective into an energy function for learning the reward distribution.

### B.3 THE FSD VIOLATION OBJECTIVE AS AN ENERGY FUNCTION

Recall the FSD-based objective in the main text:

$$\mathcal{L}_{\text{FSD}}(\pi, r) \;=\; \int_{-\infty}^{\infty} \big[F_{Z^E}(z) - F_{Z^\pi}(z)\big]_+ \, dz, \tag{21}$$

where $[x]_+ = \max\{x, 0\}$ denotes the positive part. This quantity measures, in an integrated way, how much $F_{Z^E}$ lies *above* $F_{Z^\pi}$. If $Z^E \succeq_{\text{FSD}} Z^\pi$, then $F_{Z^E}(z) \leq F_{Z^\pi}(z)$ for all $z$, so the integrand is always zero, and hence $\mathcal{L}_{\text{FSD}}(\pi, r) = 0$. If FSD is violated, then $\mathcal{L}_{\text{FSD}}(\pi, r)$ becomes positive.

**Energy-based interpretation.** We treat $\mathcal{L}_{\text{FSD}}(\pi, r)$ as an *energy* that scores how well a reward function $r$ explains the expert demonstrations under policy $\pi$. Lower $\mathcal{L}_{\text{FSD}}$ means fewer FSD violations and thus better agreement with the expert. This motivates defining an energy-based model (EBM)

$$p(\mathcal{D} \mid r) \;\propto\; \exp\big(-\mathcal{L}_{\text{FSD}}(\pi, r)\big), \tag{22}$$

where $\mathcal{D}$ denotes the expert data and the proportionality hides a (typically intractable) normalizing constant. In words: reward functions that produce small FSD violations are exponentially more likely under the expert data.

This construction gives us a *likelihood* model for the reward $r$ given the data $\mathcal{D}$, which we will combine with a prior over $r$ and then approximate via variational inference.

### B.4 VARIATIONAL INFERENCE AND ELBO DERIVATION

We now derive the variational objective used to learn the reward distribution. We start from Bayes' rule:

$$p(r \mid \mathcal{D}) \;=\; \frac{p(\mathcal{D} \mid r) \, p_0(r)}{p(\mathcal{D})},$$

where $p_0(r)$ is a prior over reward functions and

$$p(\mathcal{D}) = \int p(\mathcal{D} \mid r) \, p_0(r) \, dr$$

is the evidence (marginal likelihood), which is typically intractable to compute or differentiate.

We introduce a variational family $q_\phi(r \mid s, a)$, parameterized by $\phi$, to approximate the true posterior $p(r \mid \mathcal{D})$. To measure how close $q_\phi$ is to the true posterior, consider the KL divergence

$$\text{KL}\big(q_\phi(r \mid s, a) \,\|\, p(r \mid \mathcal{D})\big) = \mathbb{E}_{q_\phi}\Big[\log \frac{q_\phi(r \mid s, a)}{p(r \mid \mathcal{D})}\Big]. \tag{23}$$

Plugging in Bayes' rule for $p(r \mid \mathcal{D})$ gives

$$\text{KL}\big(q_\phi(r \mid s, a) \,\|\, p(r \mid \mathcal{D})\big) = \mathbb{E}_{q_\phi}\Big[\log \frac{q_\phi(r \mid s, a)}{p(\mathcal{D} \mid r) \, p_0(r)/p(\mathcal{D})}\Big] \tag{24}$$

$$= \mathbb{E}_{q_\phi}\Big[\log q_\phi(r \mid s, a) - \log p(\mathcal{D} \mid r) - \log p_0(r) + \log p(\mathcal{D})\Big]. \tag{25}$$

We can separate out the term that does not depend on $r$:

$$\text{KL}\big(q_\phi(r \mid s, a) \,\|\, p(r \mid \mathcal{D})\big) = \mathbb{E}_{q_\phi}\big[\log q_\phi(r \mid s, a) - \log p(\mathcal{D} \mid r) - \log p_0(r)\big] \;+\; \log p(\mathcal{D}). \tag{26}$$

Rearranging terms yields

$$\log p(\mathcal{D}) = \mathbb{E}_{q_\phi}\big[\log p(\mathcal{D} \mid r) + \log p_0(r) - \log q_\phi(r \mid s, a)\big] \;+\; \text{KL}\big(q_\phi(r \mid s, a) \,\|\, p(r \mid \mathcal{D})\big). \tag{27}$$

Since KL is non-negative, we obtain the *evidence lower bound* (ELBO):

$$\log p(\mathcal{D}) \geq \mathbb{E}_{q_\phi}\big[\log p(\mathcal{D} \mid r) + \log p_0(r) - \log q_\phi(r \mid s, a)\big] =: \text{ELBO}(\phi). \qquad (28)$$

Equivalently,

$$\text{ELBO}(\phi) = \mathbb{E}_{q_\phi(r \mid s,a)}\big[\log p(\mathcal{D} \mid r)\big] - \text{KL}\big(q_\phi(r \mid s, a) \,\|\, p_0(r)\big), \qquad (29)$$

which matches the expression in the main text.

**From ELBO to our reward objective.** Maximizing the ELBO is equivalent to minimizing its negative. Using the EBM likelihood from Eq. equation 22,

$$\log p(\mathcal{D} \mid r) = -\mathcal{L}_{\text{FSD}}(\pi, r) + \text{const},$$

where the constant does not depend on $r$ and thus can be dropped for optimization. Substituting into Eq. equation 29 and ignoring constants, we obtain the objective

$$\min_\phi \mathcal{L}_r(\phi) := \min_\phi \mathbb{E}_{q_\phi(r \mid s,a)}\big[\mathcal{L}_{\text{FSD}}(\pi, r)\big] + \text{KL}\big(q_\phi(r \mid s, a) \,\|\, p_0(r)\big), \qquad (30)$$

which is precisely Eq. (X) in the main text (Eq. 5 there). In other words, we learn the reward distribution by balancing two terms: (i) the expected FSD violation under $q_\phi$, and (ii) a regularization term that keeps $q_\phi$ close to the prior $p_0$.

### B.5 QUANTILES AND THE FSD LOSS

We now explain in more detail why the FSD loss in Eq. equation 21 can be expressed in terms of quantile functions, which leads to a practical way to estimate it via sampling.

**Quantile function.** For a random variable $X$ with CDF $F_X$, its (generalized) quantile function $F_X^{-1} : [0, 1] \to \mathbb{R}$ is defined by

$$F_X^{-1}(v) = \inf\{x \in \mathbb{R} \mid F_X(x) \geq v\}, \quad v \in (0, 1). \qquad$$

Intuitively, $F_X^{-1}(v)$ is the value such that a fraction $v$ of the mass of $X$ lies at or below it.

**Key identity.** We use the following identity (proved in Appendix C.1 of the main text):

$$\int_{-\infty}^{\infty} [F_{Z^E}(z) - F_{Z^\pi}(z)]_+ \, dz = \int_0^1 \big[F_{Z^\pi}^{-1}(v) - F_{Z^E}^{-1}(v)\big]_+ \, dv. \qquad (31)$$

This shows that integrating the positive difference of the CDFs is equivalent to integrating the positive difference of the *quantiles*, but with the roles of expert and policy swapped inside the bracket.

**Sketch of proof idea.** The proof relies on two facts: (i) an integral representation of the difference between two distributions in terms of their quantiles, and (ii) a change of variables between $z$ and $v$ through the CDF/quantile mapping. One can start from the left-hand side, partition the real line into regions where $F_{Z^E}(z) \geq F_{Z^\pi}(z)$ and where the opposite holds, and then perform a change of variables $z = F_{Z^\pi}^{-1}(v)$ (and similarly for the expert), carefully tracking the positive part. We refer the reader to the detailed derivation in Appendix C.1.

**Monte Carlo approximation.** The identity equation 31 is particularly useful because we can approximate quantiles from samples. For example, to approximate $F_{Z^\pi}^{-1}$, we draw $N$ return samples

$$z_n = \sum_{t=0}^{\infty} \gamma^t r_t^{(n)}, \qquad r_t^{(n)} \sim q_\phi(\cdot \mid s_t^{(n)}, a_t^{(n)}),$$

and sort them to obtain order statistics

$$z_{(1)} \leq z_{(2)} \leq \cdots \leq z_{(N)}.$$

A simple empirical approximation of the quantile function is then

$$F_{Z^\pi}^{-1}\left(\frac{k}{N}\right) \approx z_{(k)}.$$

In practice, we use such empirical quantiles (for both the expert and the learned policy) to estimate the integral on the right-hand side of Eq. equation 31 via a Riemann sum.

### B.6 Distortion Risk Measures and Their Relation to FSD

Finally, we explain how distortion risk measures (DRMs) provide a scalar, risk-sensitive summary of a return distribution and how they relate to FSD.

**Definition B.2** (Distortion function). A distortion function is a non-decreasing function $\xi : [0, 1] \to [0, 1]$ such that $\xi(0) = 0$ and $\xi(1) = 1$. Its *dual distortion* is defined as

$$\tilde{\xi}(v) := 1 - \xi(1 - v), \quad v \in [0, 1].$$

**Definition B.3** (Distortion risk measure). For an integrable random variable $X$ and a distortion function $\xi$, the associated distortion risk measure $M_\xi$ is defined by

$$M_\xi(X) \;=\; \int_0^1 F_X^{-1}(v) \, d\tilde{\xi}(v),$$

where $F_X^{-1}$ is the quantile function of $X$.

**Intuition.** The DRM $M_\xi(X)$ aggregates all quantiles of $X$ into a single scalar value, with weights determined by $d\tilde{\xi}(v)$. Different choices of $\xi$ emphasize different parts of the distribution: for example, a concave $\tilde{\xi}$ assigns more weight to *lower* quantiles, which corresponds to risk-averse behavior.

**Connection to FSD.** It is known that if $X \succeq_{\mathrm{FSD}} Y$, then

$$M_\xi(X) \;\geq\; M_\xi(Y) \quad \text{for every distortion function } \xi.$$

Furthermore, the converse holds if we require the inequality to hold for *all* distortion functions: if $M_\xi(X) \geq M_\xi(Y)$ for every distortion function $\xi$, then $X \succeq_{\mathrm{FSD}} Y$. This shows that DRMs are tightly linked to FSD: they preserve the FSD ordering if we consider all possible distortions.

In our method, we exploit this relationship by replacing the intractable indicator-based weighting of quantiles (from Eq. equation 8 in the main text) with a tractable distortion-based weighting. This yields a risk-aware policy objective of the form

$$\max_\varphi \; M_\xi(Z^{\pi_\varphi}) + \mathcal{H}(\pi_\varphi),$$

which can be optimized with standard policy gradient techniques while still encoding a meaningful notion of distributional dominance relative to the expert.

**Approximation viewpoint.** Optimizing $M_\xi(Z^{\pi_\varphi})$ for a *single* distortion function $\xi$ does not guarantee FSD dominance by itself; it corresponds to a weaker condition. However, as discussed in the main text, if one could optimize this objective for *all* distortion functions simultaneously, then under mild assumptions the resulting policy would satisfy the original FSD-based objective. Our practical objective can therefore be viewed as an approximation that focuses on a particular, user-chosen notion of risk.

## C Proofs

### C.1 Proofs for sections 4

We first wish to show that

$$\int_{-\infty}^{\infty} [F_{Z^E}(z) - F_{Z^\pi}(z)]_+ dz = \int_0^1 \left[ F_{Z^\pi}^{-1}(v) - F_{Z^E}^{-1}(v) \right]_+ dv. \tag{32}$$

**Proposition C.1.** *Let $Z^\pi$ and $Z^E$ be two real-valued integrable random variables with cumulative distribution functions $F_{Z^\pi}$ and $F_{Z^E}$, and corresponding quantile functions $F_{Z^\pi}^{-1}$ and $F_{Z^E}^{-1}$. Then we have*

$$\int_{-\infty}^{\infty} [F_{Z^E}(z) - F_{Z^\pi}(z)]_+ \, dz = \int_0^1 \left[ F_{Z^\pi}^{-1}(v) - F_{Z^E}^{-1}(v) \right]_+ \, dv,$$

*where $[x]_+ := \max(x, 0)$.*

*Proof.* Note that

$$\int_{-\infty}^{\infty} \left[ F_{Z^E}(z) - F_{Z^\pi}(z) \right]_+ dz = \int_{-\infty}^{\infty} \int_0^1 \mathbb{1}_{F_{Z^E}(z) \geq v \geq F_{Z^\pi}(z)} dv dz$$

$$= \int_0^1 \int_{-\infty}^{\infty} \mathbb{1}_{F_{Z^E}(z) \geq v \geq F_{Z^\pi}(z)} dv dz$$

$$= \int_0^1 \int_{-\infty}^{\infty} \mathbb{1}_{F_{Z^\pi}^{-1}(v) \geq z \geq F_{Z^E}^{-1}(v)} dv dz$$

$$= \int_0^1 \left[ F_{Z^\pi}^{-1}(v) - F_{Z^E}^{-1}(v) \right]_+ dv$$

The interchange of integrals are permitted by the Theorem of Fubini-Tonelli as everything is positive (Heil, 2019). Note that the definition of the quantile function (Gut & Gut, 2006) is:

$$F^{-1}(v) := \inf_{z \in \mathbb{R}} \{F(z) \geq v\}.$$

$\square$

**Proposition 4.6.** $M_\xi(X) \geq M_\xi(Y)$ *for every distortion function* $\xi$ *implies* $X \succeq_{\mathrm{FSD}} Y$.

*Proof.* Define the difference in quantile functions:

$$h(v) := F_X^{-1}(v) - F_Y^{-1}(v).$$

Suppose for contradiction that the set

$$A := \{v \in [0,1] | h(v) < 0\}$$

has positive Borel measure, i.e., $\mu(A) > 0$. Let's define a distortion function $\tilde{\xi}_A$ whose derivative is:

$$\tilde{\xi}_A'(v) = \begin{cases} \frac{1}{\mu(A)} & \text{if } v \in A, \\ 0 & \text{otherwise.} \end{cases}$$

Then $\tilde{\xi}_A$ is a valid distortion function and satisfies $\int_0^1 d\tilde{\xi}_A(v) = 1$. Note that

$$\mathcal{M}_{\xi_A}(X) - \mathcal{M}_{\xi_A}(Y) = \int_0^1 h(v) \, d\tilde{\xi}_A(v) = \int_A h(v) \cdot \frac{1}{\mu(A)} \, dv < 0.$$

This contradicts the assumption that $\mathcal{M}_{\tilde{\xi}}(X) \geq \mathcal{M}_{\tilde{\xi}}(Y)$ for all distortion functions $\tilde{\xi}$. Therefore, the set where $F_X^{-1}(v) < F_Y^{-1}(v)$ must have measure zero. Thus we have

$$F_X^{-1}(v) \geq F_Y^{-1}(v) \quad \text{for } v \in [0,1] \text{ almost everywhere (a.e.)}$$

which implies

$$F_X(z) \leq F_Y(z) \quad \text{for all } z \in \mathbb{R},$$

since

$$F_X(z) = P_X(X < z) = \mu\left(\{v \in [0,1] | F_X^{-1}(v) \leq z\}\right)$$

$$\leq \mu\left(\{v \in [0,1] \cap A^c | F_X^{-1}(v) \leq z\}\right) + \mu\left(\{v \in [0,1] \cap A | F_X^{-1}(v) \leq z\}\right)$$

$$= \mu\left(\{v \in [0,1] \cap A^c | F_X^{-1}(v) \leq z\}\right)$$

$$\leq \mu\left(\{v \in [0,1] \cap A^c | F_Y^{-1}(v) \leq z\}\right)$$

$$\leq \mu\left(\{v \in [0,1] | F_Y^{-1}(v) \leq z\}\right)$$

$$= F_Y(z)$$

The second inequality is due to the fact that for any $z$,

$$\{v \in [0,1] \cap A^c | F_X^{-1}(v) \leq z\} \subseteq \{v \in [0,1] \cap A^c | F_Y^{-1}(v) \leq z\}$$

Hence,

$$X \succeq_{\mathrm{FSD}} Y.$$

$\square$

## C.2 CONVERGENCE ANALYSIS

This appendix provides complete derivations and proofs for the convergence results summarized in Section 5. We work in the discounted MDP setting with finite action space $\mathcal{A}$ and (possibly infinite) state space $\mathcal{S}$. All function norms are $\|\cdot\|_\infty$ unless otherwise specified.

We first recall the risk–sensitive Bellman operator. For a fixed policy $\pi$, reward parameter $\phi$, and bounded $Q : \mathcal{S} \times \mathcal{A} \to \mathbb{R}$, we write

$$(\mathcal{T}_{\xi,\phi}^\pi Q)(s,a) := \mathbb{E}_\xi\big[q_\phi(s,a)\big] \;+\; \gamma\,\mathbb{E}_{\xi,\,s'\sim P(\cdot|s,a)}\big[Q(s',a')\big],$$
$$a' \sim \pi(\cdot|s'). \tag{33}$$

Here the notation $\mathbb{E}_\xi[\cdot]$ denotes the one-step evaluation combining the conditional expectation over the transition kernel and the dynamic distortion risk measure $M_\xi$ (i.e. a nested, time-consistent dynamic risk mapping). Under this formulation, $\mathcal{T}_{\xi,\phi}^\pi$ is precisely the DRM Bellman operator: it preserves the Markov structure and is a $\gamma$-contraction under mild axioms on $M_\xi$ (Ruszczyński, 2010), guaranteeing a unique fixed point $Q_{\phi,\pi}^\xi$ for each $(\phi,\pi)$.

### C.2.1 ASSUMPTIONS

We collect the standing assumptions used in the analysis.

**Assumption 5.1.** *There exists $R_{\max} < \infty$ such that*

$$|q_\phi(s,a)| \;\le\; R_{\max} \quad \textit{almost surely for all } (s,a,\phi). \tag{12}$$

This is standard in discounted RL and is enforced in our implementation by clipping the reward range (via a scaled $\tanh$ nonlinearity). It ensures that all risk-sensitive value functions are uniformly bounded.

**Assumption 5.2.** *For every $(s,a)$ and all $\phi_1, \phi_2 \in \mathbb{R}^d$, the reward laws satisfy*

$$W_\infty\big(q_{\phi_1}(\cdot|s,a),\, q_{\phi_2}(\cdot|s,a)\big) \;\le\; L_R \,\|\phi_1 - \phi_2\|, \tag{13}$$

*where $W_\infty$ denotes the Wasserstein inifity distance. Equivalently, one can couple $q_{\phi_1}(s,a)$ and $q_{\phi_2}(s,a)$ such that $|q_{\phi_1}(s,a) - q_{\phi_2}(s,a)| \le L_R\,\|\phi_1 - \phi_2\|$ almost surely.*

This assumption is mild for smooth neural parameterizations of $q_\phi(r|s,a)$ (e.g., skew-normal with smooth outputs for location, scale, and skew). It states that small changes in the reward parameters $\phi$ cannot drastically change the reward distribution, which is necessary for the critic and policy to track the moving reward model.

**Assumption 5.3.** *For each state-action pair $s \in \mathcal{S}, a \in \mathcal{A}$, the one-step distortion risk measure $M_\xi(\cdot|s,a)$ is*

1. *monotone: $X \le Y$ a.s. implies $M_\xi(X|s,a) \le M_\xi(Y|s,a)$;*

2. *translation-equivariant: $M_\xi(X + c|s,a) = M_\xi(X|s,a) + c$ for all $c \in \mathbb{R}$;*

3. *1-Lipschitz in $\|\cdot\|_\infty$: for all bounded random variables $X, Y$,*

$$\big|M_\xi(X|s,a) - M_\xi(Y|s,a)\big| \;\le\; \|X - Y\|_\infty. \tag{14}$$

For normalized distortion risk measures $M_\xi$ (including CVaR, Wang-type, and more general spectral DRMs), these properties are standard and follow from their integral representation in terms of quantile functions.

### C.2.2 CONTRACTION OF THE NESTED DRM BELLMAN OPERATOR

We now verify that $\mathcal{T}_{\xi,\phi}^\pi$ is a $\gamma$-contraction in the sup norm. This is the risk-sensitive analogue of the standard Bellman contraction and is a special instance of the general results on nested risk mappings in Ruszczyński (2010); Kopa & Šmíd (2023).

**Lemma C.2** (Contraction of $\mathcal{T}_{\xi,\phi}^{\pi}$). *Under Assumptions 5.1 and 5.3, for any fixed $(\phi, \pi)$ and any bounded $U, V : \mathcal{S} \times \mathcal{A} \to \mathbb{R}$,*

$$\left\| \mathcal{T}_{\xi,\phi}^{\pi} U - \mathcal{T}_{\xi,\phi}^{\pi} V \right\|_{\infty} \leq \gamma \left\| U - V \right\|_{\infty}. \tag{34}$$

*Proof.* For any $(s, a)$, the immediate reward terms cancel, and we have

$$\begin{aligned}
&\left| (\mathcal{T}_{\xi,\phi}^{\pi} U)(s, a) - (\mathcal{T}_{\xi,\phi}^{\pi} V)(s, a) \right| \\
&= \gamma \left| \mathbb{E}_{\xi, s' \sim P(\cdot | s, a)} [U(s', A') - V(s', A')] \right| \\
&\leq \gamma \, \mathbb{E}_{s' \sim P(\cdot | s, a)} \left[ \left| M_{\xi}(U(s', A') - V(s', A') | s') \right| \right] \\
&\leq \gamma \, \mathbb{E}_{s' \sim P(\cdot | s, a)} \left[ \| U - V \|_{\infty} \right] \; = \; \gamma \left\| U - V \right\|_{\infty},
\end{aligned} \tag{35}$$

where we used Assumption 5.3 (1-Lipschitzness) in the third line. Taking the supremum over $(s, a)$ yields 34. $\qquad\square$

By the Banach fixed-point theorem, we immediately obtain:

*Corollary C.3* (Existence and uniqueness of the risk-sensitive critic). Under Assumptions 5.1 and 5.3, for each fixed $(\phi, \pi)$ there exists a unique $Q_{\phi,\pi}^{\xi}$ solving

$$Q_{\phi,\pi}^{\xi} = \mathcal{T}_{\xi,\phi}^{\pi} Q_{\phi,\pi}^{\xi}. \tag{36}$$

Moreover, the critic is uniformly bounded.

**Lemma C.4.** *Under Assumption 5.1, let $B_Q := R_{\max}/(1 - \gamma)$. Then for all $(\phi, \pi)$,*

$$\left\| Q_{\phi,\pi}^{\xi} \right\|_{\infty} \leq B_Q. \tag{37}$$

*Proof.* By unfolding the fixed point 36 along trajectories and using $|q_{\phi}(s, a)| \leq R_{\max}$, we get for all $(s, a)$

$$\left| Q_{\phi,\pi}^{\xi}(s, a) \right| \leq \sum_{t=0}^{\infty} \gamma^t R_{\max} \; = \; \frac{R_{\max}}{1 - \gamma} \; = \; B_Q. \tag{38}$$

Taking the supremum over $(s, a)$ yields 37. $\qquad\square$

### C.2.3 SOFTMAX LIPSCHITZ PROPERTIES

We next relate $Q$-function errors to policy errors via the softmax parameterization.

**Lemma C.5.** *Let $Q, Q' : \mathcal{A} \to \mathbb{R}$ be two vectors of $Q$-values, and define*

$$\pi(a) = \frac{e^{Q(a)}}{\sum_b e^{Q(b)}}, \qquad \pi'(a) = \frac{e^{Q'(a)}}{\sum_b e^{Q'(b)}}. \tag{39}$$

*Then*

$$\| \log \pi - \log \pi' \|_{\infty} \leq 2 \| Q - Q' \|_{\infty}. \tag{40}$$

*Proof.* For any action $a$,

$$\begin{aligned}
\log \pi(a) &= Q(a) - \log \sum_b e^{Q(b)}, \\
\log \pi'(a) &= Q'(a) - \log \sum_b e^{Q'(b)}.
\end{aligned} \tag{41}$$

Subtracting,

$$\log \pi(a) - \log \pi'(a) = \big( Q(a) - Q'(a) \big) - \Big( \log \sum_b e^{Q(b)} - \log \sum_b e^{Q'(b)} \Big). \tag{42}$$

The log-sum-exp function is 1-Lipschitz in $\| \cdot \|_{\infty}$, i.e.

$$\left| \log \sum_b e^{Q(b)} - \log \sum_b e^{Q'(b)} \right| \leq \| Q - Q' \|_{\infty}. \tag{43}$$

Combining 42 and 43 gives

$$| \log \pi(a) - \log \pi'(a) | \leq | Q(a) - Q'(a) | + \| Q - Q' \|_{\infty} \leq 2 \| Q - Q' \|_{\infty}. \tag{44}$$

Taking the supremum over $a$ yields 40. $\qquad\square$

### C.2.4 LIPSCHITZ SENSITIVITY

We now show that the DRM $Q$-function depends smoothly on the reward parameters $\phi$, both for optimal control and for fixed-policy evaluation.

**Lemma C.6.** *Suppose Assumptions 5.1, 5.2, and 5.3 hold. Then for all $(s, a)$ and all $\phi_1, \phi_2$,*

$$\big| q_{\phi_1}(s,a) - q_{\phi_2}(s,a) \big| \;\leq\; L_R \, \|\phi_1 - \phi_2\|. \tag{45}$$

*Let $M_\xi$ denote the nested distortion risk functional, and assume it is 1-Lipschitz in $\|\cdot\|_\infty$ as in Assumption 5.3. Define the optimal risk-sensitive Q-function for parameter $\phi$ by*

$$Q_{\phi,*}^{\xi}(s,a) := \sup_\pi M_\xi\Big( \sum_{t=0}^{\infty} \gamma^t r_\phi(s_t, a_t) \,\Big|\, s_0 = s, a_0 = a, \pi \Big), \tag{46}$$

*where $\{(s_t, a_t)\}_{t \geq 0}$ is the trajectory under policy $\pi$ starting from $(s_0, a_0) = (s, a)$. Then there exists*

$$L_q := \frac{L_R}{1 - \gamma} \tag{47}$$

*such that for all $\phi_1, \phi_2$,*

$$\big\| Q_{\phi_1,*}^{\xi} - Q_{\phi_2,*}^{\xi} \big\|_\infty \;\leq\; L_q \, \|\phi_1 - \phi_2\|. \tag{48}$$

*Proof.* The bound on the reward smoothness is immediately due to assumption 5.2. Fix $\phi_1, \phi_2$ and $(s, a)$. For any policy $\pi$, let $\{(s_t, a_t)\}_{t \geq 0}$ be the trajectory under $\pi$ with $(s_0, a_0) = (s, a)$, and define

$$G_{\phi_i}^\pi := \sum_{t=0}^{\infty} \gamma^t q_{\phi_i}(s_t, a_t), \qquad i \in \{1, 2\}. \tag{49}$$

By definition 46,

$$Q_{\phi_i,*}^{\xi}(s,a) = \sup_\pi M_\xi\big( G_{\phi_i}^\pi \,\big|\, s, a, \pi \big), \quad i \in \{1, 2\}. \tag{50}$$

Using the inequality

$$\big| \sup_\pi f_\pi - \sup_\pi g_\pi \big| \;\leq\; \sup_\pi |f_\pi - g_\pi|, \tag{51}$$

we obtain

$$
\begin{aligned}
\big| Q_{\phi_1,*}^{\xi}(s,a) - Q_{\phi_2,*}^{\xi}(s,a) \big| &= \Big| \sup_\pi M_\xi(G_{\phi_1}^\pi | s, a, \pi) - \sup_\pi M_\xi(G_{\phi_2}^\pi | s, a, \pi) \Big| \\
&\leq \sup_\pi \big| M_\xi(G_{\phi_1}^\pi | s, a, \pi) - M_\xi(G_{\phi_2}^\pi | s, a, \pi) \big|.
\end{aligned}
\tag{52}
$$

For each fixed $\pi$, the 1-Lipschitz property of $M_\xi$ in $\|\cdot\|_\infty$ (Assumption 5.3) gives

$$
\begin{aligned}
\big| M_\xi(G_{\phi_1}^\pi | s, a, \pi) - M_\xi(G_{\phi_2}^\pi | s, a, \pi) \big| &\leq \big\| G_{\phi_1}^\pi - G_{\phi_2}^\pi \big\|_\infty \\
&= \sup_\omega \left| \sum_{t=0}^{\infty} \gamma^t \big( q_{\phi_1}(s_t(\omega), a_t(\omega)) - q_{\phi_2}(s_t(\omega), a_t(\omega)) \big) \right| \\
&\leq \sum_{t=0}^{\infty} \gamma^t \sup_{(s',a')} \big| q_{\phi_1}(s',a') - q_{\phi_2}(s',a') \big| \\
&\leq \sum_{t=0}^{\infty} \gamma^t L_R \|\phi_1 - \phi_2\| \\
&= \frac{L_R}{1 - \gamma} \, \|\phi_1 - \phi_2\|.
\end{aligned}
\tag{53}
$$

The bound does not depend on $\pi$, so combining it with Eq. 52 we obtain

$$\big| Q_{\phi_1,*}^{\xi}(s,a) - Q_{\phi_2,*}^{\xi}(s,a) \big| \;\leq\; \frac{L_R}{1 - \gamma} \, \|\phi_1 - \phi_2\|. \tag{54}$$

Taking the supremum over $(s, a)$ yields the desired result. $\qquad\square$

**Lemma C.7** (Lipschitz continuity of $Q_{\phi,\pi}^{\xi}$ in $\phi$ for fixed policy). *Suppose Assumptions 5.1, 5.2, and 5.3 hold, and fix any stationary policy $\pi$. Define the risk–sensitive evaluation Q-function as*

$$Q_{\phi,\pi}^{\xi}(s,a) := M_\xi\Big( \sum_{t=0}^{\infty} \gamma^t q_\phi(s_t,a_t) \,\Big|\, s_0=s, a_0=a, \pi \Big), \tag{55}$$

*where $\{(s_t,a_t)\}_{t\geq 0}$ is the trajectory under $\pi$ starting from $(s_0,a_0)=(s,a)$. Then for all $\phi_1, \phi_2$,*

$$\big\| Q_{\phi_1,\pi}^{\xi} - Q_{\phi_2,\pi}^{\xi} \big\|_\infty \;\leq\; L_q \,\|\phi_1 - \phi_2\|, \qquad L_q := \frac{L_R}{1-\gamma}. \tag{56}$$

*Proof.* Fix $\pi$ and $(s_0,a_0)=(s,a)$, and let $\{(s_t,a_t)\}_{t\geq 0}$ be the trajectory under $\pi$. For $i \in \{1,2\}$, define $G_{\phi_i}^{\pi}$ as in 49. Then by 55,

$$Q_{\phi_i,\pi}^{\xi}(s,a) = M_\xi(G_{\phi_i}^{\pi}|s,a,\pi), \qquad i \in \{1,2\}. \tag{57}$$

Thus

$$\begin{aligned}
\big| Q_{\phi_1,\pi}^{\xi}(s,a) - Q_{\phi_2,\pi}^{\xi}(s,a) \big| &= \big| M_\xi(G_{\phi_1}^{\pi}|s,a,\pi) - M_\xi(G_{\phi_2}^{\pi}|s,a,\pi) \big| \\
&\leq \big\| G_{\phi_1}^{\pi} - G_{\phi_2}^{\pi} \big\|_\infty \\
&\leq \frac{L_R}{1-\gamma} \,\|\phi_1 - \phi_2\|,
\end{aligned} \tag{58}$$

where the last inequality is identical to the bound in 53. Taking the supremum over $(s,a)$ gives 56. $\square$

### C.2.5 ONE-STEP CRITIC RECURSION

We now derive a simple one-step recursion for the critic's tracking error as the reward parameters $\phi_k$ and policies $\pi_k$ evolve across iterations.

For each iteration $k$, define

$$E_k := \big\| Q_{\phi_k,\pi_k}^{\xi} - Q_{\phi_k,\pi_{\phi_k}^\star}^{\xi} \big\|_\infty, \tag{59}$$

where $\pi_{\phi_k}^\star$ is an optimal DRM policy for reward parameter $\phi_k$, i.e.

$$\pi_{\phi_k}^\star \propto \mathrm{softmax}_\pi Q_{\phi_k,\pi}^{\xi}. \tag{60}$$

**Lemma C.8.** *Suppose Assumptions 5.1, 5.2, and 5.3 hold, and let $L_q$ be as in Lemma C.7. Then for all $k \geq 1$,*

$$E_k \;\leq\; \gamma E_{k-1} \;+\; 2L_q \,\|\phi_k - \phi_{k-1}\|. \tag{61}$$

*Proof.* Add and subtract $Q_{\phi_{k-1},\pi_k}^{\xi}$ and $Q_{\phi_{k-1},\pi_{\phi_{k-1}}^\star}^{\xi}$ inside the norm:

$$\begin{aligned}
&\big\| Q_{\phi_k,\pi_k}^{\xi} - Q_{\phi_k,\pi_{\phi_k}^\star}^{\xi} \big\|_\infty \\
&= \big\| Q_{\phi_k,\pi_k}^{\xi} - Q_{\phi_k,\pi_{\phi_k}^\star}^{\xi} + Q_{\phi_{k-1},\pi_k}^{\xi} - Q_{\phi_{k-1},\pi_k}^{\xi} + Q_{\phi_{k-1},\pi_{\phi_{k-1}}^\star}^{\xi} - Q_{\phi_{k-1},\pi_{\phi_{k-1}}^\star}^{\xi} \big\|_\infty \\
&\leq \big\| Q_{\phi_{k-1},\pi_{\phi_{k-1}}^\star}^{\xi} - Q_{\phi_k,\pi_{\phi_k}^\star}^{\xi} \big\|_\infty + \big\| Q_{\phi_k,\pi_k}^{\xi} - Q_{\phi_{k-1},\pi_k}^{\xi} \big\|_\infty + \big\| Q_{\phi_{k-1},\pi_k}^{\xi} - Q_{\phi_{k-1},\pi_{\phi_{k-1}}^\star}^{\xi} \big\|_\infty.
\end{aligned} \tag{62}$$

By Lemma C.6 (with $\pi_{\phi_{k-1}}^\star$ and $\pi_{\phi_k}^\star$ both optimal) and Lemma C.7 (with $\pi = \pi_k$), we have

$$\begin{aligned}
\big\| Q_{\phi_{k-1},\pi_{\phi_{k-1}}^\star}^{\xi} - Q_{\phi_k,\pi_{\phi_k}^\star}^{\xi} \big\|_\infty &\leq L_q \,\|\phi_k - \phi_{k-1}\|, \\
\big\| Q_{\phi_k,\pi_k}^{\xi} - Q_{\phi_{k-1},\pi_k}^{\xi} \big\|_\infty &\leq L_q \,\|\phi_k - \phi_{k-1}\|.
\end{aligned} \tag{63}$$

Therefore,

$$\big\| Q_{\phi_k,\pi_k}^{\xi} - Q_{\phi_k,\pi_{\phi_k}^\star}^{\xi} \big\|_\infty \;\leq\; 2L_q \,\|\phi_k - \phi_{k-1}\| \;+\; \big\| Q_{\phi_{k-1},\pi_k}^{\xi} - Q_{\phi_{k-1},\pi_{\phi_{k-1}}^\star}^{\xi} \big\|_\infty. \tag{64}$$

Next observe that for fixed $\phi_{k-1}$, $\pi^\star_{\phi_{k-1}}$ is optimal, so

$$Q^\xi_{\phi_{k-1},\pi_k} \leq Q^\xi_{\phi_{k-1},\pi^\star_{\phi_{k-1}}} \quad \text{pointwise.} \tag{65}$$

Moreover, by monotonicity of the Bellman operator and Lemma C.2,

$$0 \leq Q^\xi_{\phi_{k-1},\pi^\star_{\phi_{k-1}}} - Q^\xi_{\phi_{k-1},\pi_k} \leq \mathcal{T}^{\pi_k}_{\xi,\phi_{k-1}}\big(Q^\xi_{\phi_{k-1},\pi^\star_{\phi_{k-1}}} - Q^\xi_{\phi_{k-1},\pi_k}\big), \tag{66}$$

so taking norms and using 34 gives

$$\big\|Q^\xi_{\phi_{k-1},\pi^\star_{\phi_{k-1}}} - Q^\xi_{\phi_{k-1},\pi_k}\big\|_\infty \leq \gamma\big\|Q^\xi_{\phi_{k-1},\pi^\star_{\phi_{k-1}}} - Q^\xi_{\phi_{k-1},\pi_{k-1}}\big\|_\infty = \gamma E_{k-1}. \tag{67}$$

So that we get

$$E_k = \big\|Q^\xi_{\phi_k,\pi_k} - Q^\xi_{\phi_k,\pi^\star_{\phi_k}}\big\|_\infty \leq \gamma E_{k-1} + 2L_q\|\phi_k - \phi_{k-1}\|, \tag{68}$$

as claimed. $\qquad\square$

### C.2.6 SMOOTH REWARD UPDATES AND AVERAGED CRITIC TRACKING

We now relate the parameter drift $\|\phi_k - \phi_{k-1}\|$ to the reward update objective $\mathcal{L}_r(\phi)$ used in Eq. 5.

**Assumption C.9** (Smoothness and bounded gradients of the reward objective)**.** *Let $\mathcal{L}_r(\phi)$ denote the reward-distribution objective in Eq. 5. Assume:*

    *1. $\mathcal{L}_r$ is differentiable and its gradient is $L_\nabla$–Lipschitz:*

$$\big\|\nabla\mathcal{L}_r(\phi_1) - \nabla\mathcal{L}_r(\phi_2)\big\| \leq L_\nabla\|\phi_1 - \phi_2\| \quad \text{for all } \phi_1, \phi_2. \tag{69}$$

    *2. The iterates $\{\phi_k\}$ are projected onto a compact set $\Phi \subset \mathbb{R}^d$, so that*

$$G_{\max} := \sup_{\phi\in\Phi}\big\|\nabla\mathcal{L}_r(\phi)\big\| < \infty. \tag{70}$$

*The reward update step is*

$$\phi_k = \Pi_\Phi\big(\phi_{k-1} - \eta_{k-1}\nabla\mathcal{L}_r(\phi_{k-1})\big), \tag{71}$$

*where $\Pi_\Phi$ is the Euclidean projection onto $\Phi$ and $\{\eta_k\}$ is a deterministic stepsize schedule.*

**Lemma C.10.** *Under Assumption C.9,*

$$\|\phi_k - \phi_{k-1}\| \leq \eta_{k-1}G_{\max}. \tag{72}$$

*Proof.* By non-expansiveness of the projection,

$$\begin{aligned}\|\phi_k - \phi_{k-1}\| &= \big\|\Pi_\Phi(\phi_{k-1} - \eta_{k-1}\nabla\mathcal{L}_r(\phi_{k-1})) - \Pi_\Phi(\phi_{k-1})\big\| \\ &\leq \eta_{k-1}\big\|\nabla\mathcal{L}_r(\phi_{k-1})\big\| \leq \eta_{k-1}G_{\max},\end{aligned} \tag{73}$$

which is 72. $\qquad\square$

Now we are ready to get the main recursion formula.

**Theorem 5.4.** *Assume assumptions 5.1-5.3 hold. Let $E_k = \big\|Q^\xi_{\phi_k,\pi_k} - Q^\xi_{\phi_k,\pi^\star_{\phi_k}}\big\|_\infty$. Assume the reward update satisfies Assumption C.9, with stepsizes $\eta_k = \eta = \eta_0 K^{-\sigma}$, $\eta_0 > 0$, and $\sigma \in (0,1)$. Then running the DistIRL algorithm $K$ steps, we have*

$$\frac{1}{K}\sum_{k=1}^K E_k = \mathcal{O}(K^{-1}) + \mathcal{O}(K^{-\sigma}). \tag{15}$$

*Proof.* By Lemmas C.8 and C.10,

$$E_k \leq \gamma E_{k-1} + 2L_q G_{\max}\eta_{k-1}. \tag{74}$$

Taking the sum, we have

$$\sum_{k=1}^{K} E_k \leq \sum_{k=1}^{K} \gamma E_{k-1} + 2L_q G_{\max} \sum_{k=1}^{K} \eta_{k-1}. \tag{75}$$

Rearrange and average over $K$ gives

$$\frac{1-\gamma}{K}\sum_{k=1}^{K} E_k \leq \frac{\gamma}{K}\left(E_0 - E_K\right) + 2L_q G_{\max}\eta K^{-\sigma}. \tag{76}$$

Divide both side by $1 - \gamma$ we have

$$\frac{1}{K}\sum_{k=1}^{K} E_k \leq \frac{\gamma}{(1-\gamma)K}C_0 + \frac{1}{1-\gamma}2L_q G_{\max}\eta K^{-\sigma}. \tag{77}$$

For which we obtain the claim. $\square$

### C.2.7 POLICY CONVERGENCE IN LOG-PROBABILITY

Finally, we transfer the critic tracking guarantees to the induced policies.

**Theorem 5.5.** *For each $k$, define the learned and DRM-optimal policies induced by the current Q-functions:*

$$\pi_k(\cdot|s) \propto \exp\left(Q^\xi_{\phi_k,\pi_k}(s,\cdot)\right), \pi^\star_{\phi_k}(\cdot|s) \propto \exp\left(Q^\xi_{\phi_k,\pi^\star_{\phi_k}}(s,\cdot)\right). \tag{16}$$

*Then running the DistIRL algorithm $K$ steps, we have*

$$\frac{1}{K}\sum_{k=1}^{K}\left\|\log\pi_k - \log\pi^\star_{\phi_k}\right\|_\infty = \mathcal{O}(K^{-1}) + \mathcal{O}(K^{-\sigma}). \tag{17}$$

*Proof.* Fix $k$ and $s$. Let

$$x(\cdot) = Q^\xi_{\phi_k,\pi_k}(s,\cdot), \qquad y(\cdot) = Q^\xi_{\phi_k,\pi^\star_{\phi_k}}(s,\cdot). \tag{78}$$

By Lemma C.5,

$$\left\|\log\pi_k^+(\cdot|s) - \log\pi^{\star+}_{\phi_k}(\cdot|s)\right\|_\infty \leq 2\|x - y\|_\infty. \tag{79}$$

Taking the supremum over $s$ yields

$$\left\|\log\pi_k^+ - \log\pi^{\star+}_{\phi_k}\right\|_\infty \leq 2\left\|Q^\xi_{\phi_k,\pi_k} - Q^\xi_{\phi_k,\pi^\star_{\phi_k}}\right\|_\infty = 2E_k. \tag{80}$$

Averaging over $k = 1,\ldots,K$ and substituting the bound from Theorem 5.4 gives Eq. 17. $\square$

### C.2.8 FIRST–ORDER CONVERGENCE OF THE REWARD UPDATE

We now show that, under mild additional conditions, the reward update drives the gradient of the reward objective to zero in an averaged sense, so that the iterates approach a stationary point of the inner minimization problem over $\phi$. This is the best we can hope as we do not assume the function approximator of the reward is convex.

Recall that the reward objective $\mathcal{L}_r(\phi)$ and its update rule were introduced in Assumption C.9. The update at iteration $k$ is

$$\phi_{k+1} = \Pi_\Phi\left(\phi_k - \eta_k g_k\right), \tag{81}$$

where $g_k$ is the stochastic gradient computed using the current critic $Q^\xi_{\phi_k,\pi_k}$ and policy $\pi_k$.

**Assumption C.11** (Gradient estimator and critic bias). *Let $\mathcal{F}_k$ denote the filtration generated by all randomness up to iteration $k$. Assume that the stochastic gradient $g_k$ satisfies, for some constants $C_g, G_g > 0$,*

$$\left\| \mathbb{E}[g_k | \mathcal{F}_k] - \nabla \mathcal{L}_r(\phi_k) \right\| \leq C_g\, E_k, \tag{82}$$

$$\mathbb{E}\big[\|g_k\|^2\big] \leq G_g^2, \tag{83}$$

*where*

$$E_k := \big\| Q^\xi_{\phi_k, \pi_k} - Q^\xi_{\phi_k, \pi^\star_{\phi_k}} \big\|_\infty \tag{84}$$

*is the critic tracking error defined above.*

Intuitively, equation 82 states that the gradient bias vanishes as soon as the critic tracks the DRM–optimal $Q$ well (i.e., $E_k$ is small), which is consistent with the inequality in equation 17: a small critic gap implies a small occupancy–measure mismatch, hence a small gradient bias. The second–moment bound equation 83 is standard in nonconvex stochastic optimization.

**Theorem 5.6.** *Suppose Assumptions 5.1, 5.2, C.9, and C.11 hold. Let $\eta_k = \eta_0 k^{-\sigma}$ with $\eta_0 > 0$ and $\sigma \in (0,1)$, and assume $\mathcal{L}_r$ is bounded below on $\Phi$. Then there exists $C > 0$ such that*

$$\frac{1}{K} \sum_{k=0}^{K-1} \mathbb{E}\big[\|\nabla \mathcal{L}_r(\phi_k)\|^2\big] = \mathcal{O}\left(K^{-1}\right) + \mathcal{O}\left(K^{-\sigma}\right) + \mathcal{O}\left(K^{-1+\sigma}\right), \tag{18}$$

*Proof.* We begin from the smoothness inequality with $\phi' = \phi_{k+1}$, $\phi = \phi_k$:

$$\mathcal{L}_r(\phi_{k+1}) \leq \mathcal{L}_r(\phi_k) + \big\langle \nabla \mathcal{L}_r(\phi_k), \phi_{k+1} - \phi_k \big\rangle + \frac{L_\nabla}{2} \|\phi_{k+1} - \phi_k\|^2. \tag{85}$$

By the non-expansiveness of the projection $\Pi_\Phi$ and the update rule,

$$\|\phi_{k+1} - \phi_k\| = \big\| \Pi_\Phi(\phi_k - \eta_k g_k) - \Pi_\Phi(\phi_k) \big\|$$

$$\leq \eta_k \|g_k\|. \tag{86}$$

Moreover,

$$\big\langle \nabla \mathcal{L}_r(\phi_k), \phi_{k+1} - \phi_k \big\rangle = \big\langle \nabla \mathcal{L}_r(\phi_k), \Pi_\Phi(\phi_k - \eta_k g_k) - \phi_k \big\rangle$$

$$\leq \big\langle \nabla \mathcal{L}_r(\phi_k), -\eta_k g_k \big\rangle \tag{87}$$

$$= -\eta_k \big\langle \nabla \mathcal{L}_r(\phi_k), g_k \big\rangle. \tag{88}$$

Substituting the above into Eq. 85 yields

$$\mathcal{L}_r(\phi_{k+1}) \leq \mathcal{L}_r(\phi_k) - \eta_k \big\langle \nabla \mathcal{L}_r(\phi_k), g_k \big\rangle + \frac{L_\nabla}{2} \eta_k^2 \|g_k\|^2. \tag{89}$$

We expand the inner product using

$$\big\langle \nabla \mathcal{L}_r(\phi_k), g_k \big\rangle = \frac{1}{2}\Big( \|\nabla \mathcal{L}_r(\phi_k)\|^2 + \|g_k\|^2 - \|g_k - \nabla \mathcal{L}_r(\phi_k)\|^2 \Big),$$

which gives

$$-\eta_k \big\langle \nabla \mathcal{L}_r(\phi_k), g_k \big\rangle = -\frac{\eta_k}{2} \|\nabla \mathcal{L}_r(\phi_k)\|^2 - \frac{\eta_k}{2} \|g_k\|^2 + \frac{\eta_k}{2} \|g_k - \nabla \mathcal{L}_r(\phi_k)\|^2. \tag{90}$$

Substituting Eq. 90 into Eq. 89 we obtain

$$\mathcal{L}_r(\phi_{k+1}) \leq \mathcal{L}_r(\phi_k) - \frac{\eta_k}{2} \|\nabla \mathcal{L}_r(\phi_k)\|^2 - \frac{\eta_k}{2} \|g_k\|^2 + \frac{\eta_k}{2} \|g_k - \nabla \mathcal{L}_r(\phi_k)\|^2 + \frac{L_\nabla}{2} \eta_k^2 \|g_k\|^2$$

$$\leq \mathcal{L}_r(\phi_k) - \frac{\eta_k}{2} \|\nabla \mathcal{L}_r(\phi_k)\|^2 + \frac{\eta_k}{2} \|g_k - \nabla \mathcal{L}_r(\phi_k)\|^2 + \frac{L_\nabla}{2} \eta_k^2 \|g_k\|^2 \tag{91}$$

where we discarded the negative term $-\frac{\eta_k}{2}\|g_k\|^2$. Next we bound the bias term. Condition on $\phi_k$ and use $\|g_k - \nabla \mathcal{L}_r(\phi_k)\| \leq C_g E_k$:

$$\mathbb{E}\big[\|g_k - \nabla \mathcal{L}_r(\phi_k)\|^2\big] = \mathbb{E}\Big[\mathbb{E}\big[\|g_k - \nabla \mathcal{L}_r(\phi_k)\|^2 | \phi_k\big]\Big]$$

$$\leq \mathbb{E}\big[C_g^2 E_k^2\big]$$

$$\leq C_g^2\, \mathbb{E}[E_k^2] \leq C C_g^2\, \mathbb{E}[E_k], \tag{92}$$

where we used $E_k \geq 0$ and $E_k \leq \|Q\|_\infty$ so that $E_k^2 \leq CE_k$. Similarly,

$$\mathbb{E}\big[\|g_k\|^2\big] \leq G_{\max}^2. \tag{93}$$

Taking expectations of Eq. 91 and applying Eq. 92-93 gives

$$\mathbb{E}[\mathcal{L}_r(\phi_{k+1})] \leq \mathbb{E}[\mathcal{L}_r(\phi_k)] - \frac{\eta_k}{2}\mathbb{E}\big[\|\nabla\mathcal{L}_r(\phi_k)\|^2\big] + \frac{\eta_k}{2}C_g^2\,\mathbb{E}[E_k] + \frac{L_\nabla}{2}\eta_k^2 G_{\max}^2. \tag{94}$$

Rearrange Eq. 94 as

$$\frac{\eta_k}{2}\mathbb{E}\big[\|\nabla\mathcal{L}_r(\phi_k)\|^2\big] \leq \mathbb{E}[\mathcal{L}_r(\phi_k)] - \mathbb{E}[\mathcal{L}_r(\phi_{k+1})] + \frac{\eta_k}{2}CC_g^2\,\mathbb{E}[E_k] + \frac{L_\nabla}{2}\eta_k^2 G_{\max}^2. \tag{95}$$

Multiply both sides by $2/\eta_k$:

$$\mathbb{E}\big[\|\nabla\mathcal{L}_r(\phi_k)\|^2\big] \leq \frac{2}{\eta_k}\big(\mathbb{E}[\mathcal{L}_r(\phi_k)] - \mathbb{E}[\mathcal{L}_r(\phi_{k+1})]\big) + C_g^2\,\mathbb{E}[E_k] + L_\nabla\eta_k G_{\max}^2. \tag{96}$$

Now sum Eq. 96 over $k = 0, \ldots, K-1$:

$$\sum_{k=0}^{K-1}\mathbb{E}\big[\|\nabla\mathcal{L}_r(\phi_k)\|^2\big] \leq 2\sum_{k=0}^{K-1}\frac{\mathbb{E}[\mathcal{L}_r(\phi_k)] - \mathbb{E}[\mathcal{L}_r(\phi_{k+1})]}{\eta_k} + C_g^2\sum_{k=0}^{K-1}\mathbb{E}[E_k] + L_\nabla G_{\max}^2\sum_{k=0}^{K-1}\eta_k. \tag{97}$$

Using the boundedness equation and the fact that $\eta_k$ is constant, we bound the first sum as

$$\begin{aligned}\sum_{k=0}^{K-1}\frac{\mathbb{E}[\mathcal{L}_r(\phi_k)] - \mathbb{E}[\mathcal{L}_r(\phi_{k+1})]}{\eta_k} &= \sum_{k=0}^{K-1}\Big(\mathbb{E}[\mathcal{L}_r(\phi_k)] - \mathbb{E}[\mathcal{L}_r(\phi_{k+1})]\Big)\frac{1}{\eta_k}\\ &= \frac{K^\sigma}{\eta}\sum_{k=0}^{K-1}\Big(\mathbb{E}[\mathcal{L}_r(\phi_k)] - \mathbb{E}[\mathcal{L}_r(\phi_{k+1})]\Big)\\ &= \frac{K^\sigma}{\eta}\Big(\mathbb{E}[\mathcal{L}_r(\phi_0)] - \mathbb{E}[\mathcal{L}_r(\phi_K)]\Big)\\ &\leq \frac{\mathcal{L}_{\max} - \mathcal{L}_{\min}}{\eta}K^\sigma. \end{aligned} \tag{98}$$

Next, apply the averaged tracking bound on the action-value function:

$$\begin{aligned}\sum_{k=0}^{K-1}\mathbb{E}[E_k] &= K\cdot\frac{1}{K}\sum_{k=0}^{K-1}\mathbb{E}[E_k]\\ &= \mathcal{O}(1) + \mathcal{O}(K^{1-\sigma}) \end{aligned} \tag{99}$$

Finally, since $\eta_k = \eta K^{-\sigma}$ with $0 < \sigma < 1$,

$$\sum_{k=0}^{K-1}\eta_k = \eta\sum_{k=1}^{K}K^{-\sigma} = \mathcal{O}(K^{1-\sigma}). \tag{100}$$

Divide both sides by $K$:

$$\frac{1}{K}\sum_{k=0}^{K-1}\mathbb{E}\big[\|\nabla\mathcal{L}_r(\phi_k)\|^2\big] \leq \frac{2(\mathcal{L}_{\max} - \mathcal{L}_{\min})}{\eta}K^{1-\sigma} \tag{101}$$

$$+ \frac{\gamma}{(1-\gamma)}C_0 C_g^2 + \frac{C_g^2}{1-\gamma}2L_q G_{\max}\eta K^{1-\sigma} \tag{102}$$

$$+ L_\nabla G_{\max}^2\eta K^{1-\sigma}. \tag{103}$$

So that we have

$$\frac{1}{K}\sum_{k=0}^{K-1}\mathbb{E}\big[\|\nabla\mathcal{L}_r(\phi_k)\|^2\big] = \mathcal{O}(K^{-\sigma}) + \mathcal{O}(K^{-1+\sigma}) + \mathcal{O}(K^{-1}), \tag{104}$$

Since $0 < \sigma < 1$, all three terms vanish as $K \to \infty$, so the averaged squared gradient converges to zero. $\square$

## D MODEL ARCHITECTURE AND HYPER-PARAMETERS

Throughout this paper, we use the following model architecture for all the experiments.

Table 6: Model Parameters for DistIRL

| Parameter | Value |
|---|---|
| **Training Parameters** | |
| Learning Rate | $3 \times 10^{-4}$ |
| Batch Size | 512 |
| Total Iterations | 5,000 |
| Entropy Coefficient | 0.1 |
| Risk Measure | CVaR |
| Risk Parameter | 0.05 |
| Reward Regularization | 0.01 |
| **Network Architecture** | |
| Policy Network | [256, 128] |
| Distribution Type | Skew Gaussian |
| Reward Range | [-5.0, 5.0] |
| Number of Quantiles | 200 |
| Reward Hidden Features | 128 |

For gridworld, we specify the reward range as $[0, 2]$. For MuJoCo tasks, $[-10, 10]$. This is achieved by applying a (scaled) tanh function.

## E ADDITIONAL ABLATION STUDIES

### E.1 ABLATION ON CHOICES OF DRM AND ITS PARAMETER

In this section, we present additional ablation studies. First, we evaluate the performance of DistIRL on the risk-averse D4RL dataset with different choices of DRM in the HalfCheetah instance. Note that for CVaR and VaR, the smaller distortion parameter $\eta$ is, the more risk-averse the policy will be. But for Wang's risk measure, which has parameter $\eta$ ranging from $-1$ to $1$, the policy exihibit from risk-seeking to risk-aversion, with $\eta = 0$ having the risk-neutral behavior. The choice of risk parameter effect the shape $\tilde{\xi}'$, which affect the solution quality of the policy optimization problem in Eq. 7.

Table. 7 demonstrates the effects of different choices of risk measure and its risk parameter. Note that since the data is generated by a risk-averse policy, a risk-averse DRM produces the best result, while risk-neutral policies are substantially worse, and risk-seeking policies fail to capture the expert's behavior.

Table 7: Performance on distributional reward settings (D4RL).

| DRM | $\eta = 0.05$ | $\eta = 0.5$ | $\eta = 0.9$ | $\eta = -0.5$ | $\eta = -0.9$ |
|---|---|---|---|---|---|
| CVaR | $3539.74 \pm 44.26$ | $3384.27 \pm 151.06$ | $2851.13 \pm 689.67$ | - | - |
| VaR | $3539.12 \pm 76.77$ | $3423.43 \pm 113.72$ | $3081.96 \pm 522.94$ | - | - |
| Wang | $2670.42 \pm 730.93$ | $2849.94 \pm 1220.71$ | $3439.46 \pm 314.48$ | $1755.25 \pm 13.42$ | $444.62 \pm 1.90$ |

### E.2 ABLATION ON NUMBER OF TRAJECTORIES

Table 8: Performance averaged over 5 seeds for varying dataset sizes (10, 5, 3, 1 trajectories).

| Environment | 10 | 5 | 3 | 1 |
|---|---|---|---|---|
| HalfCheetah | $3539.74 \pm 44.26$ | $3440.67 \pm 58.48$ | $3501.53 \pm 91.82$ | $3238.49 \pm 339.72$ |
| Hopper | $886.44 \pm 0.79$ | $888.71 \pm 20.16$ | $893.15 \pm 14.13$ | $748.93 \pm 112.53$ |
| Walker2d | $1526.46 \pm 148.24$ | $1291.44 \pm 759.45$ | $1143.62 \pm 231.05$ | $1151.86 \pm 180.98$ |

In addition to the main comparison, we conduct an ablation study on the number of expert trajectories used to train our DistIRL algorithm. For each environment, we construct datasets with

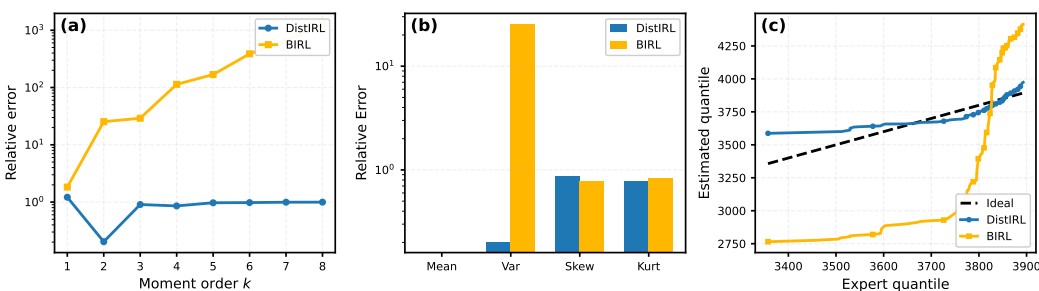

Figure 6: Caption

$\{10, 5, 3, 1\}$ expert trajectories, and train our method on each of these datasets independently. The evaluation protocol is kept identical to the main experiments. We report the average return over 5 random seeds, with the standard deviation across seeds.

Table 8 summarizes the results. Overall, the performance degrades as the number of trajectories decreases, which is expected given the reduced coverage of the expert behavior. Nevertheless, our IRL algorithm remains reasonably robust in the low-data regime. With as few as 3 to 5 trajectories, it still achieves returns close to those obtained with 10 trajectories on most tasks. Even in the extreme case of a single trajectory, the learned policies retain non-trivial performance, indicating that the method can extract useful structure from highly limited expert demonstrations.

## F    ADDITIONAL RESULTS ON MATCHING RETURN DISTRIBUTION

Figure 6 presents a comparison of distributional fidelity between DistIRL and BIRL using three metrics: (a) relative errors of higher-order moments, (b) summarized moment errors up to kurtosis, and (c) estimated–versus–expert quantile alignment. In (a), DistIRL maintains consistently low relative error across all moment orders, demonstrating its ability to capture not only the mean and variance but also the skewness and tail behavior of the expert return distribution. In contrast, BIRL's error grows rapidly with increasing moment order, indicating limited capacity to recover higher-order structure. Panel (b) further highlights this gap, showing that DistIRL achieves uniformly low errors on the first four moments, whereas BIRL exhibits substantial discrepancies, particularly in variance and higher moments. Panel (c) compares estimated and expert quantiles, where the dashed diagonal represents perfect alignment. DistIRL closely follows this ideal mapping across the entire range, while BIRL deviates significantly, especially in the upper tail. Overall, this figure illustrates that DistIRL reconstructs the full return distribution with higher accuracy than BIRL, which is necessary for risk-sensitive learning and downstream decision-making under uncertainty.

## G    ADDITIONAL RESULTS ON DOPAMINE LEVEL

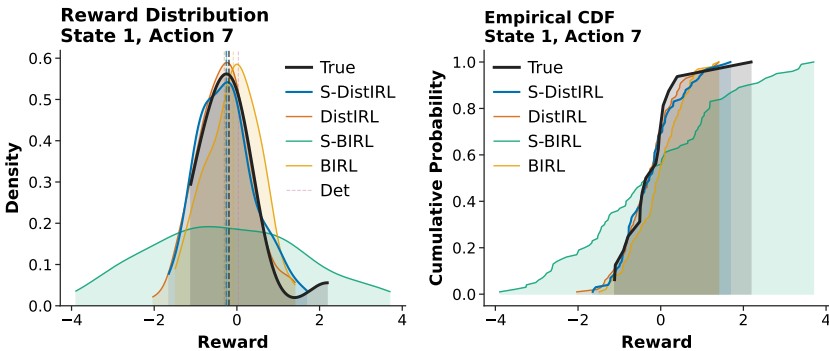

Figure 7: Reward recovery for state 1 action 7

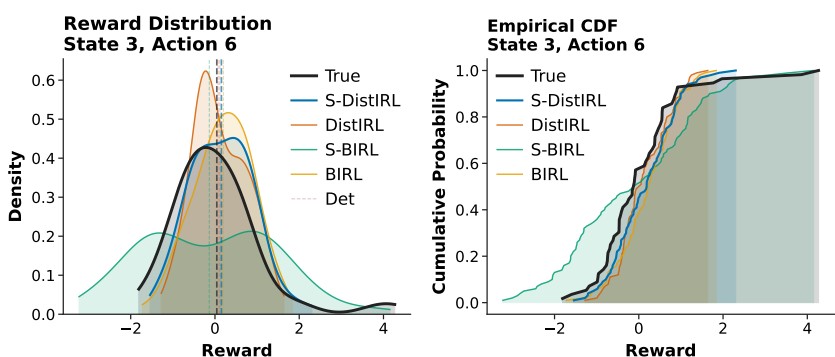

Figure 8: Reward recovery for state 3 action 6

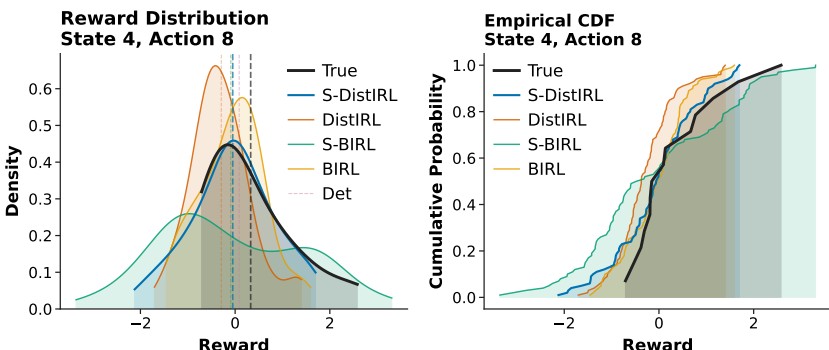

Figure 9: Reward recovery for state 4 action 8

# H  LLM USAGE AND REPRODUCIBILITY

We use LLM to aid or polish writings only. Research ideation, retrieval and discovery (e.g., finding related work) are conducted by ourselves.

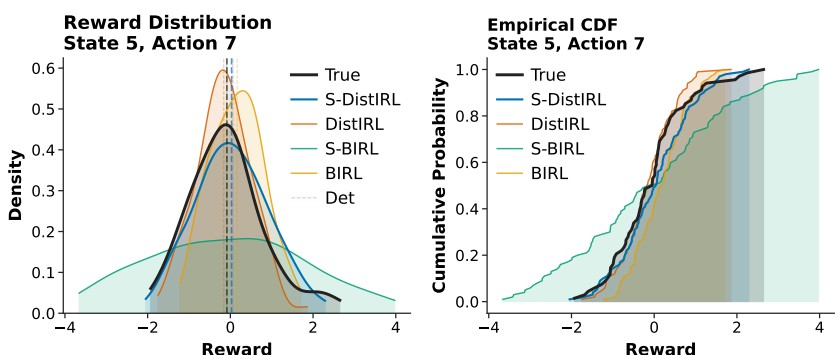

Figure 10: Reward recovery for state 5 action 7

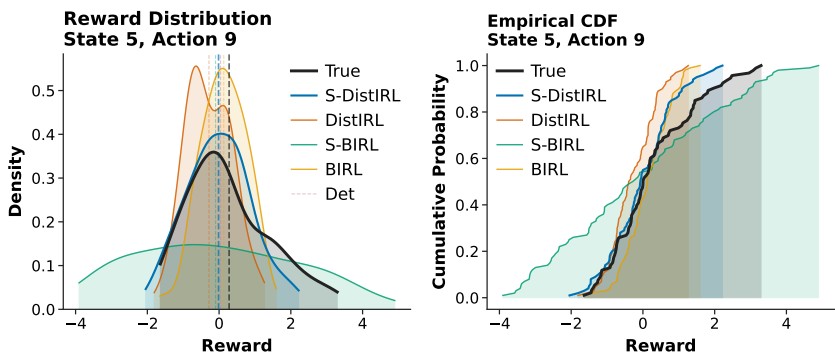

Figure 11: Reward recovery for state 5 action 9

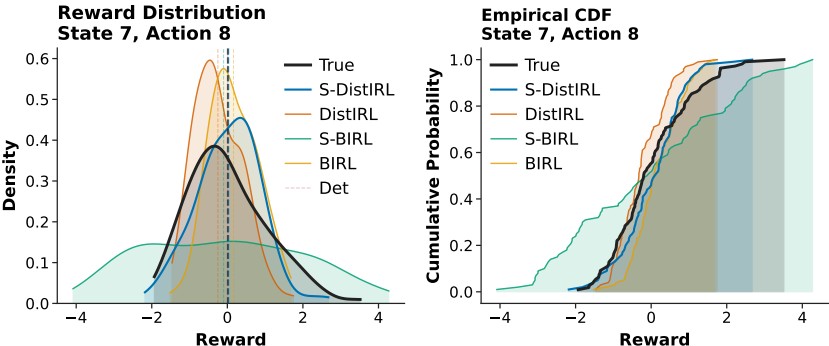

Figure 12: Reward recovery for state 7 action 8

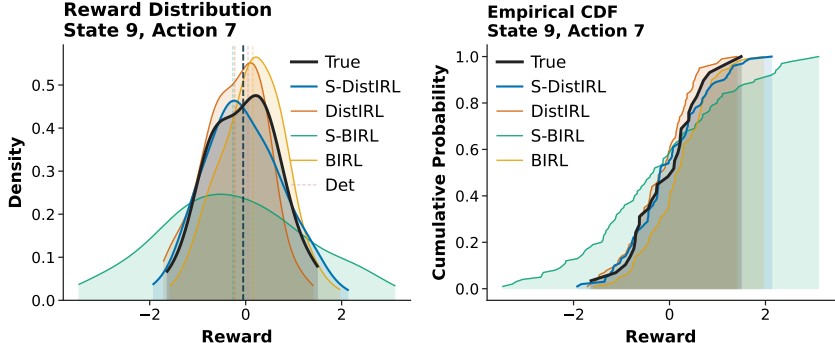

Figure 13: Reward recovery for state 9 action 7

