# OpenReview forum: "Distributional Inverse Reinforcement Learning"
_ICLR.cc/2026/Conference — Submitted to ICLR 2026_

### Official Review · Reviewer_T6Qy · 2025-10-25

**Soundness:** 4
**Presentation:** 3
**Contribution:** 3
**Rating:** 6
**Confidence:** 2

**Summary:**

This paper proposes a framework for distributional inverse reinforcement learning that jointly models the full distributions of rewards and returns instead of only their expectations. The method formulates reward recovery as minimizing violations of first-order stochastic dominance and integrates distortion risk measures to learn risk-sensitive policies under offline settings. It establishes a theoretically grounded and empirically validated approach that connects distributional reward modeling with risk-aware decision-making, extending IRL to more realistic and interpretable domains.

**Strengths:**

The paper is well written and the method is sound
The experiments are comprehensive and logically organized. The choice of baselines is appropriate, and the positioning of the proposed method is consistent.

**Weaknesses:**

The preliminaries and framework sections are somewhat lengthy and formula-heavy. Basic material such as the Bellman equation could be moved to the appendix to improve readability and keep the main text focused on the novel aspects.
The experiments do not include comparisons with the latest diffusion-based or energy-based IRL methods (from 2024–2025), which makes the empirical analysis slightly outdated.

**Questions:**

1. The method assumes that the reward follows a parameterized distribution. How valid is this assumption when applied to real neural behavioral data? If the true reward distribution is heavy-tailed or multimodal, would this lead to model bias or reduced interpretability?
2. The paper claims that DistIRL can capture higher-order reward structures. Could the authors provide concrete examples or quantitative evidence, such as skewness or kurtosis, to illustrate what is meant by “higher-order structure”?
3. The paper also states that the learned reward distribution is consistent with dopamine signals in neural data. How is this “consistency” quantitatively evaluated? Beyond Pearson correlation, are there other metrics that could better reflect nonlinear or causal relationships?

---

> ### Author Response · Authors · 2025-11-21
>
> We thank the reviewer for the encouraging words and constructive comments. Below, we refer to your comments point by point with reference to the revised manuscript.
>
> ### Weakness 1. Lengthy preliminaries and framework sections
>
> We have revised the manuscript for better readability.
>
> ### Weakness 2. Diffusion-based methods
>
> To the best of our knowledge, recent diffusion-based or EBM methods [1-3] either parameterize the critic (value functions) or the actor using diffusion models, or assume a distributional reward but use expected return as an objective. Most of which are online methods that require interactions with the environment and thus do not fall within the scope of this paper. It is possible that diffusion models can obtain a performant policy, but this does not address the unique challenge of simultaneously modeling rewards and policies. I believe the prevalent diffusion policies are equipped with scalable, accurate imitation capabilities, but their generalizability remains questionable, and perhaps reward modeling can help.
>
> [1] Lai, Chun-Mao, et al. "Diffusion-reward adversarial imitation learning." Advances in Neural Information Processing Systems 37 (2024): 95456-95487.
>
> [2] Huang, Bo-Ruei, et al. "Diffusion imitation from observation." Advances in Neural Information Processing Systems 37 (2024): 137190-137217.
>
> [3] Chen, Shang-Fu, et al. "Diffusion model-augmented behavioral cloning." arXiv preprint arXiv:2302.13335 (2023).
>
> ### Question 1. Parameterized distribution
>
> Thank you for raising this important question. The assumption of a parameterized reward distribution for the dopamine neuron example is motivated by prior findings in computational neuroscience: dopamine-related reward signals in rodents are well known to exhibit asymmetric, left-skewed variability. For this reason, we chose a skew-normal family, which captures exactly this type of asymmetric structure while remaining interpretable.
>
>
> Additionally, for prior choices, we wish to highlight that we chose skew-normal for modeling dopamine neuron response due to previous findings in computational neuroscience: dopamine-related reward signals in rodents are well known to exhibit asymmetric, left-skewed variability. The results indicate that skew-normal is indeed a better parametric family than the normal distribution, both qualitatively (Fig. 3) and quantitatively (Fig. 4).
>
> For MuJoCo tasks, however, we default to the Gaussian family, which shows competitive results across various tasks.
> We also investigate using the quantile function to parameterize the reward distribution (updated in Table 2), obtaining competitive results on the HalfCheetah instance. This opens the opportunity to use a broad range of parameterization families, including diffusion models, and we leave this as a future extension of this work.
>
> ### Question 2. Higher-order reward structures
>
> We provide additional results (in Appendix F) comparing BIRL and DistIRL on the relative errors of the actual and estimated return moments. Results show that DistIRL consistently outperforms BIRL on higher-order moments, including skewness and kurtosis.
>
> ### Question 3. Quantitative evaluation
>
> In Fig. 4b, we show the statistical distance (Wasserstein distance) between the dopamine level and the estimated reward distribution, indicating that DistIRL recovers distributions that are much more accurate than those of Bayesian IRL methods.
>
> Infering nonlinear and causal relationships is unfortunately beyond the scope of this paper, but it remains an exciting direction we would like to explore in future work.
>
>
> ### Addition of theoretical analysis
> We have added a central section (sec.~5) dedicated to the convergence analysis of our algorithm. The full proof is presented in Appendix Section C.2. The structure of the proof follows the two-timescale stochastic approximation framework. In particular, we assume we can accurately approximate the action-value function under the dynamic distortion risk measure and focus on policy and reward updates.
>
> Under specified assumptions, we show that the algorithm can converge in $ \mathcal{O}(\varepsilon^{-2}) $ steps.
> These results characterize the idealized behavior of the algorithm, thereby justifying that our approach is well aligned with concurrent first-order methods for solving IRL problems.

---

> > ### Comment · Reviewer_T6Qy · 2025-11-25
> >
> > Thanks for the response. I do not have further comments.

---

### Official Review · Reviewer_8V98 · 2025-10-27

**Soundness:** 3
**Presentation:** 3
**Contribution:** 2
**Rating:** 6
**Confidence:** 4

**Summary:**

This paper introduces a distributional framework to mitigate the stochastic challenge of offline IRL. Specifically, authors leverage variational inference framework to connect FSD based distributional optimization problem to the conventional IRL problem.

**Strengths:**

The paper is well-motivated and well-written with clear structure. The variational inference framework used to connect original IRL optimization problem with a distributional variant is interesting and elegant. The experiment section is comprehensive from low-dimensional grid world to neural science tasks and to MuJoCo locomotion tasks, validating algorithm's effectiveness. The work itself is quite complete.

**Weaknesses:**

1, The introduction and related work context should involve works on maximum causal entropy IRL based methods(MCE IRL)[1], which also target to solve the stochastic nature within the IRL problems[2,3,4,5].\
2, In Section 3.2, GAIL originally considers a MCE IRL problem instead of a MaxEnt IRL formulation.\
3, Reviewer is curious about impact of parameterizing different prior distribution. Though it has been discussed from line 314 to line 322, it would be great to have more ablation to investigate this part.
4, In addition, as an offline approach, in ablation studies it would be great to add experiments to investigate impact of trajectory quantity to algorithm performances.

[1]:Gleave, Adam, and Sam Toyer. "A primer on maximum causal entropy inverse reinforcement learning." arXiv preprint arXiv:2203.11409 (2022).\
[2]:Viano, Luca, et al. "Robust inverse reinforcement learning under transition dynamics mismatch." Advances in Neural Information Processing Systems 34 (2021): 25917-25931.\
[3]:Bloem, Michael, and Nicholas Bambos. "Infinite time horizon maximum causal entropy inverse reinforcement learning." 53rd IEEE conference on decision and control. IEEE, 2014.\
[4]:Wu, Runzhe, et al. "Diffusing states and matching scores: A new framework for imitation learning." arXiv preprint arXiv:2410.13855 (2024).\
[5]:Zhan, Simon Sinong, et al. "Model-based reward shaping for adversarial inverse reinforcement learning in stochastic environments." arXiv preprint arXiv:2410.03847 (2024).\

**Questions:**

1, The idea of stochastic reward function is interesting. However, is it equivalent to stochastic dynamic? What could be the corresponding formulation of the MDP?\
2, From the preliminaries Section 3, it seems like reward function is not stochastic? Only transition dynamic is stochastic here?\
3, What is the motivation in line 213-214 using EBM to model likelihood function with $L_{FSD}$?\

---

> ### Author Response · Authors · 2025-11-21
>
> We thank the reviewer for the encouraging words and constructive comments. Below, we refer to your comments point by point with reference to the revised manuscript.
>
> ### Weakness 1 & 2. Maximum causal entropy IRL
>
> We thank the reviewer for their feedback and have revised our manuscript to include additional MCE-based methods. Admittedly, recent work on IRL uses MCE-IRL and MaxEnt-IRL interchangeably, for which we follow the same convention.
>
> ### Weakness 3. Priors
>
> Although a standard Gaussian prior seems to be the de facto choice for Bayesian inference and generative modeling, its having a set of highly convenient mathematical and computational properties. In our context, the Gaussian prior has the maximum entropy, which fits our framework naturally. Additionally, we can easily reparameterize the distribution.
>
> We wish to highlight that we chose skew-normal for modeling dopamine neuron response due to previous findings in computational neuroscience: dopamine-related reward signals in rodents are well known to exhibit asymmetric, left-skewed variability. The results indicate that skew-normal is indeed a better parametric family than the normal distribution, both qualitatively (Fig. 3) and quantitatively (Fig. 4).
>
> For MuJoCo tasks, however, we default to the Gaussian family, which shows competitive results across various tasks.
> We also investigate using the quantile function to parameterize the reward distribution, obtaining competitive results on the HalfCheetah instance. This opens the opportunity to use a broad range of parameterization families, including diffusion models, and we leave this as a future extension of this work.
>
>
>
> ### Weakness 4. Low-data regime
>
> We provide additional ablation studies in Appendix E.2 on the effects of the number of expert trajectories, which show that DistIRL is robust even in a low-data regime.
>
> Additionally, we conduct ablation studies on the effects of various distortion risk measures and their parameter ranges, demonstrating that our algorithm requires some specification of the underlying DRM but is not overly sensitive.
>
> Admittedly, limited data can lead to overfitting. In our case, the resulting high reward for the state-action samples was observed only in the expert dataset.
>
> ### Question 1. Stochastic dynamic
>
> A stochastic reward can indeed be absorbed into the transition dynamics if one only cares about the induced return distribution and models the transition as $p(s', r|s, a)$. But to approximate this reward, we need to parameterize the reward distribution directly. Note that in this setting, for each state-action pair, its reward is a distribution, which requires its own sampling mechanism. In other words, while the MDP can be reformulated to hide the randomness inside the dynamics, learning the reward distribution still requires modeling it directly.
>
> ### Question 2. Confusion in preliminaries Section 3
>
> We have revised the manuscript to reflect the stochastic nature.
>
> ### Question 3. EBM
>
> To model the reward distribution in a principled manner, we treat $ \mathcal{L}_{\text{FSD}}(\pi, r) $
> as an energy function that scores how compatible a proposed reward $r$ is with the expert demonstrations.
>
> In particular, we define a likelihood function over the expert demonstrations $ \mathcal{D} $ using the Energy-Based Model (EBM) formulation: $ p(\mathcal{D} | r) \propto \exp\left(-\mathcal{L}_{\text{FSD}}(\pi, r)\right), $ so that reward functions that yield minor FSD violations are exponentially more likely under the expert data.   This construction is natural here because FSD does not provide an explicit probabilistic model, but \emph{does} provide a calibrated energy landscape that reflects goodness-of-fit.
> We also have a more detailed explanation in Appendix B.
>
> ### Addition of theoretical analysis
> We have added a central section (sec.~5) dedicated to the convergence analysis of our algorithm. The full proof is presented in Appendix Section C.2. The structure of the proof follows the two-timescale stochastic approximation framework. In particular, we assume we can accurately approximate the action-value function under the dynamic distortion risk measure and focus on policy and reward updates.
>
> Under specified assumptions, we show that the algorithm can converge in $ \mathcal{O}(\varepsilon^{-2}) $ steps.
> These results characterize the idealized behavior of the algorithm, thereby justifying that our approach is well aligned with concurrent first-order methods for solving IRL problems.

---

> > ### Comment · Reviewer_8V98 · 2025-11-27
> >
> > Thanks for authors effort to address my comments. At this point, I don't have any further questions, and I will maintain my score.

---

### Official Review · Reviewer_jq3Q · 2025-10-30

**Soundness:** 2
**Presentation:** 1
**Contribution:** 3
**Rating:** 4
**Confidence:** 4

**Summary:**

This paper proposes a new offline IRL approach that explicitly models stochastic reward functions and aligns full return distributions between the expert and the learned policy. Unlike conventional IRL methods that recover only deterministic reward estimates or match expected returns, the proposed approach leverages first-order stochastic dominance (FSD) as a distributional matching criterion and integrates distortion risk measures (DRMs) into policy learning to enable risk-aware imitation. The method alternatively learns a conditional reward distribution via a variational inference formulation with an energy-based likelihood derived from the FSD violation loss, and optimizes a policy that maximizes a chosen DRM over the return distribution, estimated using quantile regression.

**Strengths:**

1. The paper makes a meaningful conceptual leap from deterministic IRL to distributional IRL. Modeling a full reward distribution is a novel and valuable direction for explaining variability in expert demonstrations. The paper opens a promising avenue for studying uncertainty-aware imitation and stochastic preference modeling in decision-making.

1. The explicit use of First-Order Stochastic Dominance (FSD) as the optimization objective is innovative. Also, the formulation connects FSD, variational inference, and distortion risk measures, which is well-motivated and theoretically appealing.

1. The experiments span both synthetic and real-world (rodent behavior) domains, providing an interesting interdisciplinary case that goes beyond typical RL benchmarks.

**Weaknesses:**

1. There seems to exist theoretical flows in the derivations (correct me if I was wrong). The proof in Appendix B.1 contains an error in the change of variables involving the quantile function, incorrectly transforming the inequality $F_{Z^{\pi}}(z) \ge v$ to $z \le F_{Z^{\pi}}^{-1}(v)$ and $v \ge F_{Z^{E}}(z)$ to $F_{Z^{E}}^{-1}(v) \le z$ (should be $z \ge F_{Z^{\pi}}^{-1}(v)$ and $z \le F_{Z^{E}}^{-1}(v)$ instead), which flips the direction of the integral. This error may result in the policy objective (Eq. 9) being the negative of the theoretically correct form derived from the FSD distance, which can create a fundamental mathematical disconnect between the paper's stated FSD goal and its actual algorithmic implementation.

1. The framework's architecture combines multiple components that can be inherently prone to training instability. The minimax structure requires delicate balancing between the policy and the reward learning components to prevent divergence. This would be exacerbated by the highly complex gradient estimation required for the term $E_{q_{\phi}}[\mathcal{L}_{FSD}(\pi, r)]$.

1. The paper suffers from significant readability issues due to its immediate and simultaneous introduction of advanced concepts from disparate fields, including VI, FSD, DRM, and QR. This high conceptual load makes the core logic, especially the transformation of the complex FSD goal into the tractable maximum DRM objective for the policy, difficult to follow.

**Questions:**

1. Under what conditions is the recovered reward distribution identifiable? Could different distributions explain the same expert data?

1. In settings with very few expert trajectories (e.g., <5 or 1), how does the method perform compared to the baselines? Does the distributional reward model exhibit overfitting?

1. What if applying the learned rewards in online RL?

1. Could the authors show the convergence property of the method?

---

> ### Author Response · Authors · 2025-11-21
>
> We thank the reviewer for the encouraging words and constructive comments. Below, we refer to your comments point by point with reference to the revised manuscript.
>
> ### Weakness 1. Theoretical flows in the derivations
>
> We thank the reviewer for pointing this out. In the original manuscript, the relationship between FSD and Eq. 2 was indeed flipped, where we wish to minimize the FSD violation defined as $ \int [F_{Z^E} - F_{Z^\pi}]_+ $ (note that more minor CDF point wise implies higher probability of larger values, i.e., we want $Z^E$ dominates $Z^\pi$).
> This will lead to the same quantile relation we obtained in the paper.
> In other words, the desired relation for $Z^\pi$ and $Z^E$ leads to the FSD objective (which was incorrect in the original manuscript, but revised), which leads to the quantile objective (correct in both versions).
>
> The proof of Prop 4.6 has been amended. Intuitively, we wish to show that computing the FSD violation area (grey area in Fig. 1) by integrating over the sample space (x-axis) yields the same result as integrating over the quantile level (y-axis).
>
> We apologize for the confusion, as we derived the quantile relation first and missed the sign change. Since the implementation follows the quantile-based formulation, the results are fortunately cohesive. We have corrected the mistake in the revised manuscript.
>
> ### Weakness 2. Algorithm stability & convergence property
>
> We have added a central section (sec. 5) dedicated to the convergence analysis of our algorithm. The full proof is presented in Appendix Section C.2. The structure of the proof follows the two-timescale stochastic approximation framework. In particular, we assume we can accurately approximate the action-value function under the dynamic distortion risk measure and focus on the policy and reward updates instead.
>
> Under specified assumptions, we show that the algorithm can converge in $ \mathcal{O}(\varepsilon^{-2}) $ steps.
> These results characterize the idealized behavior of the algorithm, justifying our approach, which is well aligned with concurrent first-order methods for solving IRL problems.
>
> ### Weakness 3. Readability issues
>
> We appreciate this feedback and have significantly reorganized the draft. In particular, we
>
> a. moved the Bellman equation, QR background, and preliminaries to Appendix A.
>
> b. shortened Section 3 and added summaries before major derivations.
>
> Additionally, we have expanded the appendix with a detailed walkthrough on the necessity of the FSD objective, and its relationship to EBM, as well as a derivation of the ELBO objective under the variational inference framework in Appendix B.
>
>
> ### Question 1. Reward distribution is identifiable
>
> We thank the reviewer for the insightful question. To the best of our knowledge, the MaxEntIRL framework can identify a reward up to a global constant across actions for a given state due to reward shaping. In our case, this should translate to an identifiable, up to state-wise constant shift as well. However, since we directly compare CDFs, which fully characterize the law of a random variable, we can identify the reward distribution more strongly than its moments, as two distributions might have the same moments but different CDFs.
>
> ### Question 2. Low-data regime
>
> We provide additional ablation studies in Appendix E.2 on the effects of the number of expert trajectories, which show that DistIRL is robust even in a low-data regime.
>
> Additionally, we conduct ablation studies on the effects of various distortion risk measures and their parameter ranges, demonstrating that our algorithm requires some specification of the underlying DRM but is not overly sensitive.
>
> Admittedly, limited data can lead to overfitting. In our case, the resulting high reward for the state-action samples was observed only in the expert dataset.
>
> ### Question 3. Online RL
>
> We thank the reviewer for the comment. It's exciting to investigate the quality of the learned reward function and its transfer to the online domain, as in prior work [1-2]. The existing literature shows that the learned reward function can assist in learning robust behaviors in unseen domains. We leave this as a future extension for downstream tasks.
>
> [1] Wu, Feiyang, et al. "Infer and adapt: Bipedal locomotion reward learning from demonstrations via inverse reinforcement learning." 2024 IEEE International Conference on Robotics and Automation (ICRA). IEEE, 2024.
>
> [2] Zakka, Kevin, et al. "Xirl: Cross-embodiment inverse reinforcement learning." Conference on Robot Learning. PMLR, 2022.
>
> ### Question 4. Convergence analysis
>
> Addressed in weakness 2.

---

### Official Review · Reviewer_nY9E · 2025-10-31

**Soundness:** 2
**Presentation:** 2
**Contribution:** 2
**Rating:** 4
**Confidence:** 3

**Summary:**

This paper introduces Distributional Inverse Reinforcement Learning (DistIRL), a novel framework for offline IRL that addresses a key limitation of traditional methods: the assumption of a deterministic reward function. Conventional approaches typically recover a single point-estimate for the reward and match only the *expected* return of the expert. In contrast, DistIRL learns the full *distribution* of the reward function.

The core idea is to replace the standard mean-matching objective of MaxEntIRL with a new objective that minimizes First-order Stochastic Dominance (FSD) violations between the expert's and the learned policy's *return* distributions. This forces the model to capture higher-order moments of the return, which in turn provides a learning signal to recover the full underlying reward distribution.

To make this practical, the authors formulate the reward learning problem as a variational inference task, minimizing an FSD-based loss (Eq. 7). The corresponding policy optimization problem (Eq. 9) is intractable, but the authors propose an elegant surrogate objective (Eq. 13) based on Distortion Risk Measures (DRMs). This substitution connects the goal of distributional reward learning to the mechanism of risk-aware policy learning.

**Strengths:**

1.  **Novel and Principled Formulation:** The paper moves IRL from simple mean-matching to a more powerful distribution-matching paradigm. Using First-order Stochastic Dominance (FSD) as the objective is a principled, non-trivial, and novel approach to this problem.
2.  **Elegant Solution:** The paper provides an elegant and practical solution. It identifies the intractability of the naive policy objective and cleverly substitutes it with a tractable surrogate based on Distortion Risk Measures (DRMs). This provides a clean conceptual link: learning a reward *distribution* is dual to learning a *risk-aware* policy.
3.  **Compelling Real-World Validation:** The neuroscience experiment is a significant strength. Successfully recovering the skewed distribution of dopamine signals from raw behavioral data is a highly non-trivial result and showcases the method's potential for scientific discovery, going far beyond typical benchmark results.

**Weaknesses:**

1.  **Lack of Convergence Analysis:** The authors rightly acknowledge that "a rigorous convergence analysis is beyond the scope of this paper". While they gesture towards potential avenues for such analysis (e.g., contraction properties), the absence of any formal guarantees for the three-timescale alternating optimization algorithm (critic, policy, reward network) is a theoretical gap.
2.  **Sensitivity to DRM Choice:** The practical algorithm requires pre-selecting a specific DRM (e.g., CVaR) for the policy update. Proposition 4.6 implies that to *truly* match the FSD objective, the policy would need to be optimal under *all* DRMs, which is intractable. This raises a question: how sensitive is the recovered *reward distribution* to a "mismatch" in the chosen DRM? For example, if the expert data was generated by a risk-seeking policy, but the DistIRL algorithm uses a risk-averse CVaR objective, does it still recover the correct reward distribution? This sensitivity is not fully explored.
3.  **Minor Naming Inconsistency:** There is a confusing inconsistency in the method's name in the main results. Table 2 labels the method "IPMD (ours)", while Table 3 uses "DistIRL. (Ours)". This appears to be a typo but should be corrected for clarity.

**Questions:**

1.  Could you please clarify the naming inconsistency between Table 2 ("IPMD (ours)") and Table 3 ("DistIRL. (Ours)")? Is "IPMD" a typo, or does it refer to a specific variant?
2.  The paper's main theoretical leap is to use a specific Distortion Risk Measure (DRM) $\xi$ as a tractable surrogate for the policy objective. How sensitive is the final recovered *reward distribution* $q_{\phi}$ to this choice? Proposition 4.6 suggests that a perfect FSD match requires optimality over *all* DRMs. What happens if the chosen $\xi$ (e.g., risk-averse CVaR) does not match the true risk profile of the expert?
3.  The experiment's success hinges on using a skew-normal (S-DistIRL) parameterization for the reward. This suggests the choice of $q_{\phi}$ is important. How should a practitioner select the right parametric family for the reward distribution? Could this assumption be relaxed, for example, by using a non-parametric model for the reward distribution itself (e.g., a quantile-based representation, similar to the critic)?
4.  The paper notes the lack of a convergence analysis. Could the authors briefly elaborate on the primary technical challenges? Is it the min-max nature of the objective, the non-convexity of the neural network updates, or the stability of the three-component (critic, policy, reward) stochastic approximation?

---

> ### Author Response · Authors · 2025-11-21
>
> We thank the reviewer for the encouraging words and constructive comments. Below, we refer to your comments point by point with reference to the revised manuscript.
>
> ### Weakness 1: Lack of Convergence Analysis
>
> We have added a central section (Sec.5) dedicated to the convergence analysis of our algorithm. The full proof is presented in Appendix Section C.2. The structure of the proof follows the two-timescale stochastic approximation framework. In particular, we assume we can accurately approximate the action-value function under the dynamic distortion risk measure and focus on policy and reward updates.
>
> Under specified assumptions, we show that the algorithm can converge in $ \mathcal{O}(\varepsilon^{-2}) $ steps.
> These results characterize the idealized behavior of the algorithm, thereby justifying that our approach is well aligned with concurrent first-order methods for solving IRL problems.
>
> ### Weakness 2 Sensitivity to DRM Choice
>
> We have added ablation studies on the choices of different DRMs in Appendix E.1 and Table 7. The results show that DistIRL is not sensitive to a particular DRM, provided its risk-aversion parameter is reasonable. But suppose the data is indeed risk-averse and we choose a risk-seeking DRM. In that case, DistIRL cannot recover the optimal behavior due to the compounding error induced by misspecification in Eq. 8.
>
> Since IRL infers rewards from data under different modeling assumptions, it provides a data-driven way to test competing hypotheses about behavior (e.g., risk-averse vs.\ risk-seeking). By evaluating which model best recovers the observed optimal behavior, we can assess which hypothesis is more plausible. Suppose a version of DistIRL fails to recover the optimal behavior under a particular risk assumption. In that case, this suggests the agent or animal is unlikely to be using that assumption and instead follows a different behavioral strategy.
>
>
> ### Weakness 3. Minor Naming Inconsistency
>
> We have revised the manuscript with consistent naming. Thanks for pointing them out.
>
>
> ### Question 1. Naming Inconsistency
>
> Addressed in weakness 3.
>
> ### Question 2
>
> Addressed in weakness 2
>
> ### Question 3. Reward parametric family
>
> The choice of parametric family depends on the computational budget and specific setting for each problem.
> We wish to highlight that we chose skew-normal for modeling dopamine neuron response due to previous findings in computational neuroscience: dopamine-related reward signals in rodents are well known to exhibit asymmetric, left-skewed variability. The results indicate that skew-normal is indeed a better parametric family than the normal distribution, both qualitatively (Fig. 3) and quantitatively (Fig. 4).
>
> For MuJoCo tasks, however, we default to the Gaussian family, which shows competitive results across various tasks.
>
> We conduct an additional experiment with quantile-based parameterization on MuJoCo (updated in table 2), achieving competitive policy performance comparable to that of the DistIRL.
>
> It is an excellent point that one could use a non-parametric model for the reward distribution itself. This would indeed open up exciting opportunities for learning much more expressive reward distributions. Since our paper is the first to introduce the distributional IRL framework, we view the choice of reward parameterization as an important direction for future work.

---

> > ### Author Response · Authors · 2025-11-21
> >
> > ### Question 4
> >
> > Addressed in weakness 1.
> >
> > We'd also like to note that the convergence proof follows a min-max primal-dual first-order stochastic approximation method, specifically a two-timescale stochastic approximation scheme (TTSA), in which the outer-layer reward update converges more slowly than the inner RL problem. We do not assume convexity of the approximators, so neural nets are suitable in the analysis framework as long as the approximation is sufficiently good.

---

### Author Response · Authors · 2025-11-21
**Paper revision**

We thank the reviewer for their time and constructive comments on our paper. We have amended our manuscript with new results, both theoretical and experimental.

First, we present a convergence analysis of the proposed DistIRL algorithm with a finite-time guarantee. Under specified assumptions, we show that the algorithm can converge in $ \mathcal{O}(\varepsilon^{-2}) $ steps.
These results characterize the idealized behavior of the algorithm, justifying our approach, which is well aligned with concurrent first-order methods for solving IRL problems.

Second, we provide additional ablation studies on (1) effects of various distortion risk measures as well as their parameter range, showcasing that our algorithm requires some level of specification of the underlying DRM, but not overly sensitive; (2) effects of the number of trajectories of the expert data, which shows that DistIRL is robust even in a low-data regime.

Additionally, for prior choices, we wish to highlight that we chose skew-normal for modeling dopamine neuron response due to previous findings in computational neuroscience: dopamine-related reward signals in rodents are well known to exhibit asymmetric, left-skewed variability. The results indicate that skew-normal is indeed a better parametric family than the normal distribution, both qualitatively (Fig. 3) and quantitatively (Fig. 4).

For MuJoCo tasks, however, we default to the Gaussian family, which shows competitive results across various tasks.
We also investigate using the quantile function to parameterize the reward distribution, obtaining competitive results on the HalfCheetah instance. This opens the opportunity to use a broad range of parameterization families, including diffusion models, and we leave this as a future extension of this work.

We have updated our manuscript and reflected the changes in blue texts.

Thanks again to all the reviewers for dedicating their time to the feedback.

---

### Meta-Review · Area_Chair_CVhW · 2026-01-06

**Summary:**

This paper introduces Distributional Inverse Reinforcement Learning (DistIRL), which aims to recover full reward distributions and learn risk-aware policies by matching return distributions via first-order stochastic dominance, supported by theoretical analysis and experiments across synthetic tasks, neuroscience data, and MuJoCo benchmarks.

Reviewers agreed the idea is novel and well-motivated, with strong empirical coverage. However, they raised recurring concerns about theoretical correctness and clarity, including potential derivation errors, strong or unclear assumptions in the convergence analysis, sensitivity to the choice of distortion risk measures and reward parameterization, and training stability. Presentation and readability issues further complicated assessment.

Although the rebuttal addressed several points and added analysis, key concerns about soundness, robustness, and clarity remain. Considering the reviews, discussion, and response, the paper does not yet reach the required level of confidence for acceptance.

**Reviewer Concerns:**

Please see my summary.

**Reviewer Scores:**

It is difficult to say.  Overall, the authors provided some solid rebuttal, but it's a subjective judgement for the reviewer whether they would like to raise their score.

---

### Decision · Program_Chairs · 2026-01-26

Reject